# Maximizing utility in multi-agent environments by anticipating the behavior of other learners

**Angelos Assos**
MIT CSAIL
Cambridge, MA
assos@mit.edu

**Yuval Dagan**
UC Berkeley
Berkeley, CA
yuvaldag@berkeley.edu

**Constantinos Daskalakis**
MIT CSAIL
Cambridge, MA
costis@mit.edu

## Abstract

Learning algorithms are often used to make decisions in sequential decision-making environments. In multi-agent settings, the decisions of each agent can affect the utilities/losses of the other agents. Therefore, if an agent is good at anticipating the behavior of the other agents, in particular how they will make decisions in each round as a function of their experience that far, it could try to judiciously make its own decisions over the rounds of the interaction so as to influence the other agents to behave in a way that ultimately benefits its own utility. In this paper, we study repeated two-player games involving two types of agents: a learner, which employs an online learning algorithm to choose its strategy in each round; and an optimizer, which knows the learner's utility function and the learner's online learning algorithm. The optimizer wants to plan ahead to maximize its own utility, while taking into account the learner's behavior. We provide two results: a positive result for repeated zero-sum games and a negative result for repeated general-sum games. Our positive result is an algorithm for the optimizer, which exactly maximizes its utility against a learner that plays the Replicator Dynamics — the continuous-time analogue of Multiplicative Weights Update (MWU). Additionally, we use this result to provide an algorithm for the optimizer against MWU, i.e. for the discrete-time setting, which guarantees an average utility for the optimizer that is higher than the value of the one-shot game. Our negative result shows that, unless P=NP, there is no Fully Polynomial Time Approximation Scheme (FPTAS) for maximizing the utility of an optimizer against a learner that best-responds to the history in each round. Yet, this still leaves open the question of whether there exists a polynomial-time algorithm that optimizes the utility up to $o(T)$.

## 1 Introduction

With the increased use of learning algorithms as a means to optimize agents' objectives in unknown or complex environments, it is inevitable that they will be used in multi-agent settings as well. This encompasses scenarios where multiple agents are repeatedly taking actions in the same environment, and their actions influence the payoffs of the other players. For example, participants in repeated online auctions use learning algorithms to bid, and the outcome, who gets to win and how much they pay, depends on all the bids (see e.g. [39, 40]). Other examples arise in contract design (see e.g. [30]),

38th Conference on Neural Information Processing Systems (NeurIPS 2024).

Bayesian persuasion (see e.g. [19]), competing retailers that use learning algorithms to set their prices (see e.g. [25, 14]), and more.

It is thus natural to contemplate the following: in a setting where multiple agents take actions repeatedly, using their past observations of the other players' actions to decide on their future actions, what is the optimal strategy for an agent? This question is rather difficult, hence players often resort to simple online learning algorithms such as *mean-based* learners: those algorithms select actions that approximately maximize the performance on the past, and include methods such as MWU, Best Response Dynamics, FTRL, FTPL, etc; see e.g. [16]. Yet, when an agent knows that the other agents are using mean-based learners, it can it can adjust its strategy to take advantage of this knowledge. This was shown in specific games [9, 24]; however, in general settings it is not known how to best play against mean-based learners, even if the agent knows *everything* about the learners, and can predict with certainty what actions they would take as a function of the history of play. This raises the following question:

**Meta Question.** *In a multi-agent environment where agents repeatedly take actions, what is the best strategy for an agent, if it knows that the other agents are using mean-based learners to select their actions? Can it plan ahead if it can predict how the other agents will react? Can such an optimal strategy be computed efficiently?*

**Setting.** We study environments with two agents: a *learner* who uses a learning algorithm to decide on its actions, and an *optimizer*, that plans its actions, by trying to optimize its own reward, taking into account the learner's behavior. The setting is modeled as a repeated game: in each iteration $t = 1, \ldots, T$, the optimizer selects a strategy $x(t)$, which is a distribution over a finite set of *pure actions* $\mathcal{A} = \{a_1, \ldots, a_n\}$, i.e. $x(t) \in \Delta(\mathcal{A})$. At the same time, the learner selects a distribution $y(t)$ over $\mathcal{B} = \{b_1, \ldots, b_m\}$, i.e. $y(t) \in \Delta(\mathcal{B})$. The strategies $x(t)$ and $y(t)$ are viewed as elements of $\mathbb{R}^n$ and $\mathbb{R}^m$, respectively, and the elements of $\mathcal{A}$ and $\mathcal{B}$ are identified with unit vectors. In each round, each player gains a utility, which is a function of the strategies played by both agents. The utilities for the optimizer and the learner equal $x(t)^\top A y(t)$ and $x(t)^\top B y(t)$, respectively, where $A, B \in \mathbb{R}^{n \times m}$. The goal of each player is to maximize their own utility, which is summed over all rounds $t = 1, \ldots, T$. We split our following discussion according to the study of zero-sum and general-sum games, as defined below.

**Zero-sum games.** *Zero-sum* games are those where $A = -B$. Namely, the learner and the optimizer have opposing goals. Such games have a *value*, which determines the best utility that the players can hope for, if they are both playing optimally. It is defined as:

$$\mathrm{Val}(A) = \max_{x \in \Delta(\mathcal{A})} \min_{y \in \Delta(\mathcal{B})} x^\top A y = \min_{y \in \Delta(\mathcal{B})} \max_{x \in \Delta(\mathcal{A})} x^\top A y \,.$$

The two quantities in the definition above are equal, as follows from von Neumann's minmax theorem. The first of these quantities implies that the optimizer has a strategy that guarantees them a utility of $\mathrm{Val}(A)$ against any learner, and the second quantity implies that the learner has a strategy which guarantees that the optimizer's utility is at most $\mathrm{Val}(A)$. Such strategies, namely, those that guarantee minmax value, are termed *minmax strategies*.

If the learner would have used a minmax strategy in each round $t$, then the optimizer could have only received a utility of $T \mathrm{Val}(A)$. Yet, when the learner uses a learning algorithm, the optimizer can receive higher utility. A standard theoretical guarantee in online learning is *no-regret*. When a learner uses a *no-regret* algorithm to play a repeated game, it is guaranteed that the optimizer's reward after $T$ rounds is at most $T \mathrm{Val}(A) + o(T)$. Yet, $T$ is finite, and the $o(T)$-term can be significant. Consequently, we pose the following question, which we study in Section 2.

**Question 1.** *In repeated zero-sum games between an optimizer and a learner, when can the optimizer capitalize on the sub-optimality of the learner's strategy, to obtain significantly higher utility than the value of the game? Can the optimizer's algorithm be computationally efficient?*

**General-sum games.** Games where $B \neq -A$ are termed *general-sum*. These games are significantly more intricate: they do not posses a value or minmax strategies. In order to study such games, various notions of *game equilibria* have been proposed, such as the celebrated Nash Equilibrium. In the context related to the topic of our paper, if the optimizer could commit on a strategy, and the other player would best respond to it, rather than using a learning algorithm, then the optimizer could

get the *Stackelberg* value, by playing according to the *Stackelberg Equilibrium*. Further, there is a polynomial-time algorithm to compute the optimizer's strategy in this case [21, 41, 20].

Against a learner that uses a no-regret learning algorithm, the optimizer could still guarantee the Stackelberg value of the game, by playing according to the Stackelberg Equilibrium. Yet, it was shown that in certain games, the optimizer can gain up to $\Omega(T)$ more utility compared to the Stackelberg value when they are playing against mean-based learners [9, 24, 37, 30, 18]. However, this often requires playing a carefully-planned strategy which changes across time. While such a strategy could be computed in polynomial-time for specific games [9, 13, 23], no efficient algorithm was known for general games. Deng et al. [24] devised a control problem whose solution is approximately the optimal strategy for the optimizer, against mean-based learners. Yet, they do not provide a computationally-efficient algorithm for this control problem. Finding an optimal strategy for the learner was posed as an open problem [24, 30, 11], that can be summarized as follows:

**Question 2.** *In repeated general-sum games between an optimizer and a mean-based learner, is there a polynomial-time algorithm to find an optimal strategy for the optimizer?*

We study this question in Section 3.

## 1.1  Our results

Our results are two-fold; In zero sum games, we provide positive results, and show what the optimizer's optimal rewards and strategies for a given game should be, in both discrete and continuous time games. Our results in that realm are the following:

- For continuous time games, we provide an exact, closed form solution of the rewards of the optimizer against a learner that uses the replicator dynamics, i.e. continuous MWU (Section 2, Theorem 1). In the same theorem we prove that the optimizer can achieve optimal rewards by playing a constant strategy throughout the game. i.e. $x(t) = x^*, x \in \Delta(\mathcal{A}) \forall t \in [0, T]$.

- We also prove a range where the optimal rewards might be and provide a lower bound for when $\eta T \to \infty$ (Section 2, Proposition 1).

- For discrete time games, when the optimizer is up against a learner that uses MWU, we first prove that the optimal rewards of the optimizer in this setting, will be lower bounded by the rewards of the optimizer in the continuous time game, if they use the same strategy $x^*$ (Section 2 Proposition 2).

- We then prove that the reward 'gap' between continuous and discrete time games cannot be greater than $\eta T/2$ (Section B, Proposition 10), and there are games that achieve a gap of at least $\Omega(\eta T)$ (Section 2, Proposition 3). In fact we give a class of games that can achieve that gap in rewards (Section 2,Proposition 4).

In general-sum games, we provide the first known computational lower bound for calculating the optimal strategy against a mean based learner. We formalize the problem of optimizing rewards as a decision problem, where the answer for an instance is 'YES' if the optimizer can achieve rewards more than $T$ and 'NO' if the optimizer can receive rewards at most $T - 1$. We prove that there is no polynomial time algorithm, assuming $P \neq NP$ that distinguishes between the two cases by using a reduction from Hamiltonian Cycle. The formal theorem and a sketch of the reduction construction can be found in Section 2 Theorem 4.

## 1.2  Related Work

The game-theoretic modeling of multi-agent environments has long history [16, 45, 27]. Such studies often revolve around proving that if multiple agents use learning algorithms to repeatedly play against each other, the avarege history of play converges to various notions of game equilibria [43, 16]. Recent works have shown that some learning algorithms yield fast convergence to game equilibria and fast-decaying regret, if all players are using the same algorithm [44, 22, 2, 4, 26, 42, 46].

**Optimizing against no regret learners.**  Braverman et al. [9] initiated a recent thread of works, which studies how a player can take advantage of the learning algorithms run by other agents in order to gain high utility. They showed that in various repeated auctions where a seller sells an item to a single buyer in each iteration $t = 1, \ldots, T$, the seller can gain nearly the maximal utility that they

could possibly hope for, if the learner runs an EXP3 learning algorithm (i.e., the seller's utility can be arbitrarily close to the total welfare). This can be $\Omega(T)$ larger than what they can get if the buyer is strategic. Deng et al. [24] generalized the study to arbitrary normal-form games. They showed an example for a game where an optimizer that plays against any mean-based learner gets $\Omega(T)$ more than the Stackelberg value, which is what they would get against strategic agents. The same paper further showed that against no-swap-regret learners, the optimizer cannot gain $\Omega(T)$ more than the Stackelberg value and Mansour et al. [37] showed that no-swap-regret learners are the only algorithms with this property. Brown et al. [11] showed a polynomial time algorithm to compute the optimal strategy for the optimizer against the no-regret learner that is most favorable to the optimizer — yet, such no-regret learner may be unnatural. Additional work obtained deep insights into the optimizer/learner interactions in general games [11, 5], in Bayesian games [37] and in specific games such as auctions [23, 13, 35, 34], contract design [30] and Bayesian persuasion [18].

**Optimizing against MWU in** $2 \times 2$**-games.** Guo and Mu [29] obtain a computationally-efficient algorithm for an optimizer that is playing a zero-sum game against a learner that uses MWU, in games where each player holds 2 actions, and they analyze that optimal strategy.

**Regret lower bounds for online learners.** Regret lower bounds for learning algorithms are tightly related to upper bounds for an optimizer in zero-sum games. The optimizer can be viewed as an adversary that seeks to maximize the regret of the algorithm. The main difference is that these lower bounds construct worst-case instances, yet, we seek to maximize the utility of the optimizer in any game. Regret lower bounds for MWU and generalizations of it were obtained by [17, 32, 15, 28]. Minmax regret lower bounds against any online learner can be found, for example, in [16].

**Approximating the Stackelberg value.** Lastly, the problem of computing the Stackelberg equilibrium is well studied [21, 41, 20]. Recent works also study repeated games between an optimizer and a learner, where the optimizer does not know the utilities of the learner and its goal is to learn to be nearly as good as the Stackalberg strategy [6, 38, 36, 31].

## 2  Optimizing utility in zero-sum games

**Continuous-time games.** We begin our discussion with continuous-time dynamics, where $x(t)$ and $y(t)$ are functions of the continuous-time parameter $t \in [0, T]$. The total reward for the optimizer is $\int_0^T x(t)^\top A y(t) dt$, whereas the learner's utility equals the optimizer's utility multiplied by $-1$. We assume that the learner is playing the *Replicator Dynamics* with parameter $\eta > 0$, which is the continuous-time analogue of the Multiplicative Weights Update algorithm, which plays at every time $t$:

$$y_i(t) = \frac{\exp\left(\eta \int_0^t x(s)^\top B e_i ds\right)}{\sum_{j=1}^m \exp\left(\eta \int_0^t x(s)^\top B e_j ds\right)} , \quad i = 1, \ldots, m . \tag{1}$$

This formula gives a higher weight to actions that would have obtained higher utility in the past. When $\eta$ is larger, the learner is more likely to play the actions that maximize the past utility. The value of $\eta$ usually becomes smaller as $T$ increases, and typically $\eta \to 0$ as $T \to \infty$.

The following definition will be helpful for the analysis:

**Definition 1** (Historical Rewards for the learner)**.** *The historical rewards for the learner in continuous games at time $t$, denoted by $h(t) \in \mathbb{R}^m$, is an $m$ dimensional vector where $h_i(t), i = 1, \ldots, m$ corresponds to the sum of rewards achieved by action $a_i$ of the learner against the history of play in the game so far. We assume a general setting where the learner at $t = 0$ might have non-zero historical rewards, i.e. $h(0) \in \mathbb{R}^m$. If the strategy of the optimizer is $x : [0, t] \to \Delta(\mathcal{A})$ we get:*

$$h(t) = h(0) + \int_0^t B^\top x(t) dt$$

Suppose the game has been played for some time $t$ and the learner has collected historical rewards $h(t)$. The total reward that the optimizer gains from the remainder of this game can be written as a

function of the time left for the game $T - t$, the historical rewards of the learner at time $t$, $h(t)$, and the strategy $x : [0, T - t] \to \Delta(\mathcal{A})$ of the optimizer for the remainder of the game.

$$R_{cont}(x, h(t), T - t, A, B) = \int_0^{T-t} \frac{\sum_{i=1}^m e^{\eta\left(h_i(u) + e_i^\top B^\top \int_0^u x(s)ds\right)} \cdot e_i^\top A^\top x(u)}{\sum_{i=1}^m e^{\eta\left(h_i(u) + e_i^\top B^\top \int_0^u x(s)ds\right)}} du \qquad (2)$$

For simplicity, we can just transfer the time to $0$ and assume that the historical rewards of the learner are just $h(0)$. That way we can rewrite the above definition as:

$$R_{cont}(x, h(0), \tau, A, B) = \int_0^\tau \frac{\sum_{i=1}^m e^{\eta\left(h_i(t) + e_i^\top B^\top \int_0^t x(s)ds\right)} \cdot e_i^\top A^\top x(t)}{\sum_{i=1}^m e^{\eta\left(h_i(t) + e_i^\top B^\top \int_0^t x(s)ds\right)}} dt \qquad (3)$$

In general we are interested in finding the maximum value that the optimizer can achieve at any moment of the game:

$$R_{cont}^*(h(0), \tau, A, B) = \max_x R_{cont}(x, h(0), \tau, A, B) \qquad (4)$$

For finding the optimal reward for the optimizer from the beginning of the game we would have to find $R_{cont}^*(\mathbf{0}, T, A, B)$, where $\mathbf{0} = (0, 0, \dots, 0)^\top$.

The next theorem characterizes the exact optimal strategy for the optimizer, against a learner that uses the Replicator Dynamics in zero sum games:

**Theorem 1.** *In a zero-sum continuous game, when the learner is using the Replicator Dynamics, the optimal rewards for the optimizer can be achieved with a constant strategy throughout the game, i.e. $x(t) = x^* \in \Delta(\mathcal{A})$. The optimal reward is obtained by the following formula:*

$$R_{cont}^*(h(0), T, A, -A) = \max_{x \in \Delta(\mathcal{A})} \left\{ \frac{\ln\left(\sum_{i=1}^m e^{\eta h_i(0)}\right) - \ln\left(\sum_{i=1}^m e^{\eta\left(h_i(0) - T e_i^\top A^\top x\right)}\right)}{\eta} \right\} \qquad (5)$$

*Further, $x^*$ is the maximizer in the formula above.*

*Proof.* For zero sum games in continuous time, we have that equation 3 becomes:

$$R_{cont}(x, h(0), T, A, -A) = \int_0^T \frac{\sum_{i=1}^m e^{\eta\left(h_i(0) - e_i^\top A^\top \int_0^t x(s)ds\right)} \cdot e_i^\top A^\top x(t)}{\sum_{i=1}^m e^{\eta\left(h_i(0) - e_i^\top A^\top \int_0^t x(s)ds\right)}} dt \qquad (6)$$

Notice that

$$\frac{d}{dt}\left(e^{\eta\left(h_i(0) - e_i^\top A^\top \int_0^t x(s)ds\right)}\right) = e^{\eta\left(h_i(0) - e_i^\top A^\top \int_0^t x(s)ds\right)} \cdot \frac{d}{dt}\left(\eta h_i(0) - \eta \cdot e_i^\top A^\top \int_0^t x(s)ds\right)$$

From Leibniz rule we have that $\frac{d}{dt} \int_0^t x(s)ds = x(t)$, thus we get:

$$\frac{d}{dt}\left(e^{\eta\left(h_i(0) - e_i^\top A^\top \int_0^t x(s)ds\right)}\right) = -e^{\eta\left(h_i(0) - e_i^\top A^\top \int_0^t x(s)ds\right)} \cdot \eta \cdot e_i^\top A^\top x(t)$$

Given that, note that we can find a closed form solution for $R_{cont}(x, h(0), T, A, -A)$ as follows:

$$R_{cont}(x, h(0), T, A, -A) = \left[-\frac{1}{\eta} \ln\left(\sum_{i=1}^m e^{\eta\left(h_i(0) - e_i^\top A^\top \int_0^t x(s)ds\right)}\right)\right]_0^T$$

$$= \frac{\ln\left(\sum_{i=1}^m e^{\eta h_i(0)}\right) - \ln\left(\sum_{i=1}^m e^{\eta\left(h_i(0) - e_i^\top A^\top \int_0^T x(s)ds\right)}\right)}{\eta}$$

Suppose the optimal rewards $R_{cont}^*(h(0), T, A, -A)$ are achieved with $x^{opt}(t)$. Note that the final reward only depends on $\int_0^T x(s)ds$, thus the same reward that is achieved by $x^{opt}(t)$, can be achieved by $x^* = \frac{1}{T} \int_0^T x^{opt}(s)ds$. Thus there exists a $x^*$ such that:

$$R_{cont}^*(h(0), T, A, -A) = \frac{\ln\left(\sum_{i=1}^m e^{\eta h_i(0)}\right) - \ln\left(\sum_{i=1}^m e^{\eta\left(h_i(0) - e_i^\top A^\top x^* T\right)}\right)}{\eta} \qquad (7)$$

which is what we wanted. $\qquad \square$

The optimal strategy $x^*$ for the optimizer in continuous zero-sum games can be therefore obtained by finding the minimum of the convex function $f(x) = \ln\left(\sum_{i=1}^m e^{\eta\left(h_i(0) - e_i^\top A^\top x T\right)}\right)$. We can compute the optimal strategy of the optimizer in an efficient way using techniques from convex optimization. More details can be found in the appendix at Proposition 5.

We further analyze how larger the optimal achievable reward is, compared to the naive bound of $T\operatorname{Val}(A)$. We specifically show that always the optimal rewards of the optimizer are in the range of $[T\operatorname{Val}(A), T\operatorname{Val}(A) + \log(m)/\eta]$, where $m$ is the number of actions of the learner, showing the optimizer can always get more utility than just the value of the game.

We also analyze what happens to the rewards of the optimizer as $\eta T \to \infty$. First, let us define for any optimizer's strategy $x$, the set of all best-responses,

$$\operatorname{BR}(x) = \arg\max_{b \in \mathcal{B}} x^\top B b.$$

This defines a set, and if there are multiple maximizers, $|\operatorname{BR}(x)| > 1$. We then have the following proposition:

**Proposition 1** (informal)**.** *The optimal reward for an optimizer playing against a learner that uses the replicator dynamics with parameter $\eta$ in an $n \times m$ game, is in the range $[T\operatorname{Val}(A), T\operatorname{Val}(A) + \log(m)/\eta]$. Further, as $\eta T \to \infty$, this optimal utility is at least*

$$T\operatorname{Val}(A) + \frac{\log(m/k)}{\eta}, \qquad \text{where } k = \min_{x \in \Delta(\mathcal{A}) \text{ minmax strategy}} |\operatorname{BR}(x)|.$$

The proof of Proposition 1 can be found in Appendix B, Propositions 6 and 7. We note that the limiting utility is obtained by playing constantly any minmax strategy $x$ that attains the minimum in the definition of $k$ above.

**Connection to optimal control and the Hamilton-Jacobi-Bellman equation.** We will present another way of achieving Theorem 1, using literature from control theory. One can view the problem of maximizing the optimizer's utilities as an optimal control problem; what control (or strategy) should the optimizer use in order to maximize their utility given that the learner has some specific dynamics that depend on the control of the optimizer? The Hamilton-Jacobi-Bellman equation [8] gives us a partial differential equation (PDE) of $R_{cont}^*(h, t, A, B)$ that if we solve, we can find a closed form solution of the optimal utility of the optimizer. The equation (for general sum games):

$$-\frac{dR_{cont}^*(h, t, A, B)}{dt} = \max_{x \in \Delta(\mathcal{A})} \left( \frac{\sum_{i=1}^m e^{\eta h_i} \cdot e_i^\top A^\top x}{\sum_{i=1}^m e^{\eta h_i}} + (\nabla_h R_{cont}^*(h, t, A, B))^\top \cdot B^\top x \right)$$

The intuition of the PDE is as follows; the current state of the learner can be defined given only the history $h$ of the sum of what the optimizer played so far, and the time left in the game $t$. Given we are at a state $h, t$, the optimal rewards for the optimizer are going to be equal to the rewards of playing action $x \in \Delta(\mathcal{A})$ for time $\Delta t$ added together with the optimal reward in the new state, namely $R_{cont}^*(h + B^\top x \Delta t, t + \Delta t, A, B)$. Taking the limit as $\Delta t \to 0$, gives us the above partial differential equation.

Plugging in $B = -A$, for the case of zero-sum games, we get:

$$-\frac{dR_{cont}^*(h, t, A, -A)}{dt} = \max_{x \in \Delta(\mathcal{A})} \left( \frac{\sum_{i=1}^m e^{\eta h_i} \cdot e_i^\top A^\top x}{\sum_{i=1}^m e^{\eta h_i}} - (\nabla_h R_{cont}^*(h, t, A, -A))^\top \cdot A^\top x \right)$$

If one plugs in the formula we calculated in Theorem 1, they would find that indeed it is a solution to the above PDE.

**Discrete-time games.** We now move to the discrete-time setting. The learner is assumed to be playing the Multiplicative-Weights update algorithm, defined by:

$$y_i(t) = \frac{\exp\left(\eta \sum_{s=0}^{t-1} x(s)^\top B e_i\right)}{\sum_{j=1}^m \exp\left(\eta \sum_{s=1}^{t-1} x(s)^\top B e_j\right)}, \quad i = 1, 2, \ldots, m. \tag{8}$$

We show that if the learner constantly plays the strategy $x^*$ that is optimal for continuous-time, the obtained utility against MWU can only be higher, compared to playing against the Replicator Dynamics in continuous-time, with the same step size $\eta$:

**Proposition 2** (informal). *Let $A$ be a zero-sum game matrix and $\eta > 0$ and $T \in \mathbb{N}$. Let $x^* \in \Delta(\mathcal{A})$ be the optimal strategy against the replicator dynamics, from Theorem 1. Then, the utility in the discrete time, achieved by an optimizer which plays $x(t) = x^*$ for all $t \in \{1, \ldots, T\}$ against MWU with parameter $\eta$, is at least the utility achieved in the continuous-time by playing $x^*$ against the replicator dynamics with the same parameters $\eta, T$.*

The proof can be found in Appendix B (Proposition 8).

Proposition 2 implies that Proposition 1 provides lower bounds on the achievable utility of this constant discrete-time strategy. In order to further analyze its performance, we would like to ask how much larger the discrete-time utility of the optimizer can be from the optimal continuous-time utility. We include the following statement, which follows from a standard approach:

**Proposition 3.** *Let $T \in \mathbb{N}$ and $\eta > 0$. For any zero-sum game whose utilities are bounded in $[-1, 1]$, the best discrete-time optimizer obtains a utility of at most $\eta T/2$ more than the best continuous-time optimizer. Further, there exists a game where the best discrete-time optimizer achieves a utility of $\tanh(\eta)T/2 = (\eta - O(\eta^2))T/2$ more than the optimal continuous-time optimizer, and $\tanh(\eta)T/2$ more than the discrete-time optimizer from Proposition 2.*

The proof can be found in Appendix B (Propositions 9 and 10).

Proposition 3 implies that in some cases, the best discrete-time optimizer can gain $T \operatorname{Val}(A) + \Omega(\eta T)$. We would like to understand in which games this is possible for any choice of $\eta \in (0, 1)$. We provide a definition that guarantees that this would be possible:

**Condition 1.** *There exists a minmax strategy $x$ for the optimizer such that there exist two best responses for the learner, $b_{i_1}, b_{i_2} \in \operatorname{BR}(x)$, which do not coincide on $\operatorname{support}(x)$. Namely, there exists an action $a_k \in \operatorname{support}(x)$ such that $a_k^\top A b_{i_1} \neq a_k^\top A b_{i_2}$.*

To motivate Condition 1, notice that in order to achieve a gain of $\Omega(\eta T)$ for any $\eta$, the discrete optimizer has to be significantly better than the continuous-time optimizer. For that to happen the learner would have to change actions frequently. Indeed, the difference between the discrete and continuous learners is that the discrete learner is slower to change actions: they only change actions at integer times, whereas the continuous learner could change actions at each real-valued time. In order to change actions, they need to have at least two good actions, and this is what Assumption 1 guarantees. We derive the following statement:

**Proposition 4.** *For any zero-sum game $A \in \mathbb{R}^{n \times m}$ that satisfies Assumption 1, $\eta \in (0, 1)$ and $T \in \mathbb{N}$, there exists an optimizer, such that against a learner that uses MWU with step size $\eta$, achieves a utility of at least $T \operatorname{Val}(A) + \Omega(\eta T)$, where the constant possibly depends on $A$.*

The proof can be found in Appendix B (Proposition 11).

Proposition 4 considers a strategy for the optimizer which takes the minmax strategy $x$ from Assumption 1, and splits into two strategies: $x', x'' \in \Delta(\mathcal{A})$, such that $(x' + x'')/2 = x$. Further, $x'^\top A b_{i_1} \neq x''^\top A b_{i_2}$, where $b_{i_1}, b_{i_2}$ are defined in Assumption 1. It plays $x'$ in each odd round $t$, and $x''$ in each even round. It is possible to show that in each two consecutive iterations, the reward for the optimizer is $2 \operatorname{Val}(A) + \Omega(\eta)$. Summing over $T/2$ consecutive pairs yields the final bound.

## 3 A computational lower bound for optimization in general-sum games

In this section, we present the first limitation on optimizing against a mean-based learner. Specifically, we study the algorithm which is termed *Best-Response* or *Fictitious Play* [10, 43]. At each time step $t = 1, \ldots, T$, this algorithm selects an action $y(t)$ that maximizes the cumulative utility for rounds $1, \ldots, t-1$:

$$y(t) = \arg \max_{y \in \Delta(\mathcal{B})} \sum_{s=1}^{t-1} x(s)^\top B y \tag{9}$$

There is always a maximizer which corresponds to a pure action $b_i \in \mathcal{B}$ of the learner and we assume that if there are ties, they are broken lexicographically. In this section, we constrain the optimizer to also play pure actions.

To put this algorithm in context, we note the following connections to general mean-based learners: (1) *mean-based* learners are any algorithms which select an action $y(t)$ that approximately maximizes Eq. (9); (2) Best-Response is equivalent to MWU with $\eta \to \infty$.

We prove that there is no algorithm that approximates the optimal utility of the optimizer up to an approximation factor of $1 - \epsilon$, whose runtime is polynomial in $n, m, T$ and $1/\epsilon$, as exemplified by the following (informal) Theorem:

**Theorem 2** (informal). *Let* Alg *be an algorithm that receives parameters $\epsilon > 0$, $m, n, T \in \mathbb{N}$ and utility matrices $A, B$ of dimension $m \times n$ and entries in $[0, 1]$, and outputs a control sequence $x(1), \dots, x(T) \in \mathcal{A}$. Let $U$ denote the utility attained by the learner and let $U^*$ denote the optimal possible utility. If $U \geq (1 - \epsilon)U^*$ for any instance of the problem, and if $P \neq NP$, then* Alg *is not polynomial-time in $m, n, T$ and $1/\epsilon$.*

A sketch of the proof can be found below, and a full proof can be found in the Appendix C. The proof is obtained via a reduction from the Hamiltonian cycle problem. We note two limitations of this result: (1) The lower bound is for $T = n/2 + 1$, and it shows hardness in distinguishing between the case that the optimal reward is $T$ and the case that the optimal reward is at most $T - 1$. It is still open whether one could efficiently find a sequence that optimizes the reward up to $o(T)$; (2) fictitious-play is a fundamental online learning algorithm. Yet, it does not possess the no-regret guarantee. It is still open whether one could obtain maximal utility against no-regret learners such as MWU with a small step size.

We continue by framing the problem of maximizing rewards against a Best-Response learner as a decision problem, called OCDP:

**Problem 1** (Optimal Control Discrete Pure (OCDP)). *An OCDP instance is defined by $(A, B, n, m, k, T)$, where $n, m, k, T \in \mathbb{N}$, $A \in \{0, 1\}^{n \times m}$ and $B \in [0, 1]^{n \times m}$. The numbers $n$ and $m$ correspond to the actions of the optimizer and learner in a game, where $A$ is the utility matrix of the optimizer and $B$ is the utility matrix of the learner. This instance is a 'YES' instance if the optimizer can achieve utility at least $k$ after playing the game for $T$ rounds with a learner that uses the Best Response Algorithm (Eq. (9)), and 'NO' if otherwise.*

We will prove that OCDP is NP-hard, using a reduction from the Hamiltonian cycle problem.

**Problem 2** (Hamiltonian cycle). *Given a directed graph $G(V, E)$, find whether there exists a Hamiltonian cycle, i.e. a cycle that starts from any vertex, visits every vertex exactly once and closes at the same vertex where it started.*

It is a known theorem that the Hamiltonian Cycle is an NP-complete problem, as it is one of Karp's 21 initial NP-Complete problems.

**Theorem 3** ([33]). *Hamiltonian Cycle is NP-complete.*

We conclude with the main result of this section, followed by a proof sketch. The full proof appears in Section C.

**Theorem 4.** *OCDP is NP-hard. That is, if $P \neq NP$, there exists no algorithm that runs in polynomial time in $n, m$ and $k$ which distinguishes between the case that a reward of $k$ is achievable and the case that it is impossible to obtain reward more than $k - 1$.*

*Proof sketch.* Consider an instance of the Hamiltonian cycle problem: $\pi_H = (V, E)$, where $V = \{v_1, \dots, v_n\}$, $E = \{e_1, \dots, e_m\}$. We create an instance of OCDP as follows. First, set $T = k = n + 1$. We will construct the instance such that the optimizer can receive reward $n + 1$ if and only if there is a Hamiltonian cycle in the graph. Define the optimizer's actions to be $\{a_1, \dots, a_m\}$, and the learner's actions to be $\{b_1, \dots, b_n, b'_1, \dots, b'_n\}$. Namely, for each node $v_i$ of the graph, the learner has two associated actions, $b_i$ and $b'_i$. The details in the sketch differ slightly from the proof for clarity. The reduction is constructed to satisfy the following properties:

- The only way for the optimizer to receive maximal utility, is to play a strategy as follows: the first $n$ actions should correspond to edges $e_{i_1} - e_{i_2} - \cdots - e_{i_n}$ that form a Hamiltonian cycle which starts and ends at the vertex $v_1$; and the last action corresponds to $e_{i_{n+1}}$ which is an outgoing edge from $v_1$.

- We define the utility matrix for the learner such that, if the optimizer is playing according to this strategy, then the learner's actions will correspond to the vertices of the same cycle, denoted as $v_{j_1} - \cdots - v_{j_n} - v_{j_1}$. This is achieved by defining the learner's tie-breaking rule to play $a_1$ in the first round; and defining the learner's utilities such that, if the optimizer has played in rounds $1, \ldots, t-1$ the edges along a path $v_{j_1} - \cdots - v_{j_t}$, then the best response for the learner at time $t$ would be to play $a_{j_t}$. To achieve this, we define for any edge $e_k = (v_p, v_q)$: $B[a_k, b_p] = -1$, $B[a_k, b_q] = 1$ and $B[a_k, b] = 0$ for any other action $b$. Consequently, after observing the edges along $b_{j_1} - \cdots - b_{j_t}$, the cumulative reward for the learner would be 1 for $b_{j_t}$, $-1$ for $b_{j_1}$ and 0 for any other $b_j$. Consequently, the learner will best respond with $b_{j_t}$ — we ignore the actions $b'_1, \ldots, b'_n$ of the learner at the moment.

- In order to guarantee that the above optimizer's strategy yields a utility of $n+1$, we define the optimizer's utility such that $A[e_k, b_p] = 1$ if $e_k = (v_p, v_q)$ and for all $q \neq p$ $A[e_k, b_q] = 0$. This will also force the optimizer to play edges that form a path. Indeed, if the optimizer has previously played the edges along a path $v_{j_1} - \cdots - v_{j_t}$, then, as we discussed, the learner will play $b_{i_t}$ in the next round. For the optimizer to gain a reward at the next round, they must play an edge outgoing from $a_{j_t}$. This preserves the property that the optimizer's actions form a path.

- Next, we would like to guarantee that the optimizer's actions align with a Hamiltonian cycle. Therefore, we need to ensure that the path played by the optimizer does not close too early. Namely, if the learner has played the edges along the path $v_{j_1} - \cdots - v_{j_t}$ and if $t < n$, then $j_1, \ldots, j_t$ are all distinct. For this purpose, the actions $v'_1, \ldots, v'_n$ are defined. We define for any edge $e_k = (v_i, v_j)$: $B[a_k, b'_i] = 0.85$. This will prevent the optimizer from playing the same vertex twice, for any round $t = 1, \ldots, n$ (recall that there are $T = n+1$ rounds), as argued below.

  Assume for the sake of contradiction that the first time the same vertex is visited is at $t$, where $j_t = j_r$ for some $r < t \leq n$. Then, the cumulative utility of action $b'_{j_r}$ for rounds $1, \ldots, t$ is 1.7, which is larger than any other action. This implies that at the next round, the learner will play action $b'_{j_r}$. We define the utility for the optimizer to be zero against any action $b'_j$. Hence, any scenario where the learner plays an action $b'_j$, prevents the optimizer from receiving a utility of $n+1$. This happens whenever $i_t = i_j$ for some $j < t \leq n$. Hence, an optimizer that receives a reward of 1 at any round must play a path that does not visit the same vertex twice, in rounds $t = 1, \ldots, n$.

- Lastly, notice that we do want the optimizer to play the same vertex $a_1$ twice, at rounds 1 and $n+1$. This does not prevent the optimizer from receiving optimal utility. Indeed, if the optimizer would play the action $a_1$ twice, then the learner would play $b'_1$ at the next round. Yet, since there are only $n+1$ rounds, there is no "next round" and no reward is lost. The details that force the learner to play $b_1$ in round $n+1$ appear in the full version of the proof.

The above explanation sketches why it is possible to get $n+1$ reward if and only if there is a Hamiltonian cycle and this concludes the reduction. $\square$

## 4 Conclusion and future directions

In this paper we studied how an optimizer should play in a two-player repeated game knowing that the other agent is a learner that is using a known mean-based algorithm. In zero-sum games we showed how they can gain optimal utility against the Replicator Dynamics and we further analyzed the utility that they could gain against MWU. In general sum games, we showed the first computational hardness result on optimizing against a mean-based learner, by reduction from Hamiltonian Cycle.

One interesting problem that remains is the open is analyzing the optimal reward in general sum games against the Replicator Dynamics (or the MWU), which was denoted as $R^*_{cont}(\mathbf{0}, T, A, B)$. In the fully general case with no restriction on the utility matrices $A$ and $B$, we believe there is no closed form solution for the optimal utility, differently from the zero-sum case. However, it would be interesting to understand how $R^*_{cont}(\mathbf{0}, T, A, B)$ behaves as a function of $A$ and $B$ and how the best strategy for the optimizer looks like. A conjecture in this direction was given by Deng et al. [24]. Perhaps it would be easier to study simpler scenarios, such as the one where $\text{rank}(A + B) = 1$,

which has been explored in the context of computing equilibria and the convergence of learning algorithms to them ([1], [3]). Another direction is improving on the lower bound for general sum games. Currently, we prove that it is hard to distinguish between the case where the optimizer can achieve reward $\alpha = T$ and the case where the optimizer cannot achieve more than $\beta = T - 1$. Is it also hard to distinguish between a reward of at least $\alpha$ or at most $\beta$ in cases where $\alpha - \beta = \Omega(T)$? Are there lower bounds when the learner uses different learning algorithms, such as MWU? Other relevant open directions are extensions to multi-agent settings [13], analyzing how the learner's utility is impacted by interaction with the optimizer in general-sum games [30], which learning algorithms yield higher utilities against an optimizer, and what algorithms should be used to both learn and optimize at the same time?

**Societal impact.** The work studies multi-agent environments and how agents can benefit by anticipating the behavior of other agents. We believe that increasing the academic knowledge in this topic can help learning agents assess their risk of being utilized by other agents and can help to build algorithms that are more resilient to manipulation. As always with new technologies, there is a risk that malicious players will utilize ideas from this paper, which could cause a harmful effect on other agents.

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

# A  Additional Definitions

We introduce the following definitions that will help us in the future proofs.

**Definition 2** (Historical Rewards for the learner). *The historical rewards for the learner in continuous games at time $t$, denoted by $h(t) \in \mathbb{R}^m$, is an $m$ dimensional vector that corresponds to the sum of rewards achieved by each action of the learner against the history of play in the game so far. We assume a general setting where the learner at $t = 0$ might have non-zero historical rewards, i.e. $h(0) \in \mathbb{R}^m$. If the strategy of the optimizer is $x : [0, t] \to \Delta(\mathcal{A})$ we get:*

$$h(t) = h(0) + \int_0^t B^\top x(t) dt$$

*In the discrete setting, given that the optimizer has played $x_0, x_1, \ldots, x_{t-1} \in \Delta(\mathcal{A})$, we have for $t \geq 1$:*

$$h(t) = h(0) + \sum_{i=0}^{t-1} B^\top x_i$$

**Definition 3** (Value of the game). *In a zero sum case, where $B = -A$ we define the value of the game for the optimizer as $\mathrm{Val}(A)$:*

$$\mathrm{Val}(A) = \min_{y \in \Delta(\mathcal{B})} \max_{x \in \Delta(\mathcal{A})} x^\top A y$$

**Definition 4** (Best Response Set). *For a given strategy of the optimizer $x \in \Delta(\mathcal{A})$, denote by $\mathrm{BR}(x)$ the set of best responses by the learner:*

$$\mathrm{BR}(x) = \left\{ e_i | i \in \mathrm{argmin}_{i \in [m]} \left[ x^\top A e_i \right] \right\}$$

**Definition 5** (Min-max strategies). *Denote with $\mathrm{MinMaxStrats}_{opt}(A)$ the set of strategies of the optimizer that achieve the value of the game.*

$$\mathrm{MinMaxStrats}_{opt}(A) = \left\{ x \in \Delta(\mathcal{A}) \ \middle| \ \mathrm{Val}(A) = \min_{i \in [m]} x^\top A e_i \right\}$$

## A.1  Algorithms for the learner

For completeness, we provide pseudocode for the algorithms.

**Definition 6** (MWU Algorithm). *Fix the step size $0 \leq \eta \leq \frac{1}{2}$. In a game where the utility matrix of the learner is $B$, the MWU algorithm is as follows:*

---
**Algorithm 1** MWU algorithm
---
1: **procedure** MWU$(B, T, \eta)$
2:     $h(t) = (0, 0, \ldots, 0)^\top$
3:     **for** $t = 1, 2, \ldots, T$ **do**
4:         $y \leftarrow \left( \frac{e^{\eta \cdot h_1(t)}}{\sum_{i=1}^m e^{\eta \cdot h_i(t)}}, \ldots, \frac{e^{\eta \cdot h_m(t)}}{\sum_{i=1}^m e^{\eta \cdot h_i(t)}} \right)$
5:         Submit $y$
6:         Observe $x_t$
7:         $h(t+1) = h(t) + B^\top x$
8:     **end for**
9: **end procedure**

---

**Definition 7** (Replicator Dynamics). *Suppose $x : [0, T] \to \Delta(\mathcal{A})$ is a strategy for the optimizer. If the learner is using the replicator dynamics, the strategy for the learner at time $t$ is given by:*

$$y_i(t) = \frac{\exp\left( \eta \int_0^t x(s)^\top B e_i \right)}{\sum_{j=1}^n \exp\left( \eta \int_0^t x(s)^\top B e_j \right)}, i = 1, 2, \ldots, m \,. \tag{10}$$

**Definition 8** (Best Response Algorithm). *In a game where the utility matrix of the learner is $B$, the best response algorithm is as follows, assuming we break ties lexicographically (i.e. if $h_i(t) = h_j(t)$, then $b_i$ beats $b_j$ if $i < j$):*

---

**Algorithm 2** Best Response algorithm

---

1: **procedure** BRA($B, T$)
2: $\quad h(0) = (0, 0, \ldots, 0)^\top$
3: $\quad$ **for** $t = 0, 1, \ldots, T - 1$ **do**
4: $\quad\quad y \leftarrow b_{\mathrm{argmax}_{i \in [m]} h_i(t)}$
5: $\quad\quad$ Submit $y$
6: $\quad\quad$ Observe $x_t$
7: $\quad\quad h(t + 1) = h(t) + B^\top x$
8: $\quad$ **end for**
9: **end procedure**

---

# B  Optimization in zero-sum games: missing proofs

In this section, we present some ommited proofs of lemmas mentioned in the main section.

## B.1  Continuous games

First, we prove that the optimal strategy for the optimizer in continuous games can be computed efficiently. From theorem 1 we know that the optimal strategy for the optimizer throughout the game, is just $x(t) = x^*$, where $x^*$ minimizes $f(x)$:

$$f(x) = \ln \left( \sum_{i=1}^{m} e^{\eta \left( h_i(0) - T e_i^\top A^\top x \right)} \right)$$

The above is also known as the log-sum-exp function. This is a convex function, which allows us to compute its minima efficiently. We use the following two lemmas.

**Lemma 1** (Example 5.15 [7]). *The log-sum-exp function $f : \mathbb{R}^n \to \mathbb{R}$, for which $f(\mathbf{x}) = \ln \left( \sum_{i=1}^{m} e^{x_i} \right)$, is 1-smooth with respect to the $\ell_2, \ell_\infty$ norms and convex.*

**Lemma 2** (Theorem 3.8, [12]). *For a function $f : \mathcal{X} \to \mathbb{R}$ that is convex and $\beta$-smooth with respect to a norm $\|\cdot\|$ and where $R = \sup_{x,y} \|x - y\|$, the Frank-Wolfe algorithm can find $x_t$ on $O(t)$ time for which:*

$$f(x_t) - f(x^*) \le \frac{2\beta R^2}{t + 1}$$

We can therefore compute the approximate optimal strategy for the optimizer efficiently:

**Proposition 5.** *For zero sum continuous games where the learner is using replicator dynamics, we can find an $\epsilon$ approximate optimal strategy for the optimizer in $O(\frac{1}{\epsilon\eta})$ time. That is we can find $x$ for which:*

$$R^*_{cont}(\mathbf{0}, T, A, -A) - R_{cont}(x, \mathbf{0}, T, A, -A) \le \epsilon$$

*Proof.* Consider the function $g(x) = \ln \left( \sum_{i=1}^{m} e^{\eta \left( h_i(0) - e_i^\top A^\top x T \right)} \right)$. From 1, we also know that $g$ is $1-$smooth with respect to the $\ell_\infty$ norm. Note that since the strategies are on the simplex, we have that $R = \sup_{x,y} \|x - y\| = 1$. Using lemma 2, with $t = \frac{2}{\epsilon\eta} - 1$ we can find $x$ for which $g(x) - g(x^*) \le \epsilon\eta$. Using theorem 1, we get:

$$R^*_{cont}(\mathbf{0}, T, A, -A) - R_{cont}(x, \mathbf{0}, T, A, -A) = \frac{g(x) - g(x^*)}{\eta} \le \epsilon$$

as required. $\qquad\square$

Next, we bound the range of the possibilities for the utility gained by the best optimizer:

**Proposition 6.** *In the zero-sum continuous setting, the optimizer's optimal utility is bounded as follows:*

$$\text{Val}(A) \cdot T \le R^*_{cont}(\mathbf{0}, T, A, -A) \le \text{Val}(A) \cdot T + \frac{\ln m}{\eta} \tag{11}$$

*where* $\text{Val}(A)$ *is as defined in 3.*

*Proof.* We have:

$$
\begin{aligned}
R^*_{cont}(\mathbf{0}, T, A, -A) &= \frac{1}{\eta}\left(\ln(m) - \min_x \ln\left(\sum_{i=1}^m e^{-\eta \cdot e_i^\top A^\top xT}\right)\right) = \\
&= \frac{1}{\eta}\left(\ln(m) - \min_{x\in\Delta(\mathcal{A})} \max_{j\in\{1,2,\ldots,m\}} \ln\left(e^{-\eta e_j^\top A^\top xT}\sum_{i=1}^m e^{\eta(e_j^\top - e_i^\top)A^\top xT}\right)\right) = \\
&= \frac{1}{\eta}\left(\ln(m) - \min_{x\in\Delta(\mathcal{A})} \max_{j\in\{1,2,\ldots,m\}}\left(-\eta e_j^\top A^\top xT + \ln\left(\sum_{i=1}^m e^{\eta(e_j^\top - e_i^\top)A^\top xT}\right)\right)\right) = \\
&= \frac{1}{\eta}\left(\ln(m) + \max_{x\in\Delta(\mathcal{A})} \min_{j\in\{1,2,\ldots,m\}}\left(\eta e_j^\top A^\top xT - \ln\left(\sum_{i=1}^m e^{\eta(e_j^\top - e_i^\top)A^\top xT}\right)\right)\right)
\end{aligned}
$$

Note that picking out the $j$ for which $e_j^\top A^\top x \le e_i^\top A^\top x$ for all $i \in [m]$, makes $e^{\eta(e_j^\top - e_i^\top)A^\top xT} \le 1$ for all $i = 1, 2, \ldots, m$. On the other hand $e^{\eta(e_j^\top - e_j^\top)A^\top xT} = 1$ . That gives $\ln 1 = 0 \le \ln\left(\sum_{i=1}^m e^{\eta(e_j^\top - e_i^\top)A^\top xT}\right) \le \ln m$. We finally have:

$$\max_{x\in\Delta(\mathcal{A})} \min_{j\in\{1,2,\ldots,m\}} e_j^\top A^\top xT \le R^*_{cont}(\mathbf{0}, T, A, -A) \le \max_{x\in\Delta(\mathcal{A})} \min_{j\in\{1,2,\ldots,m\}} e_j^\top A^\top xT + \frac{\ln m}{\eta} \tag{12}$$

which is exactly what we want. $\square$

If we take the limit of the game as $\eta T$ goes to infinity, we can lower bound the optimal rewards of the optimizer by considering a strategy from the set $\text{MinMaxStrats}(A)$ as defined in 5, that has the least number of best responses.

**Proposition 7.** *As $\eta T \to \infty$,*

$$
\begin{aligned}
&R^*_{cont}(\mathbf{0}, T, A, -A) \\
&\ge \text{Val}(A) \cdot T + \frac{1}{\eta}\left(\ln(m) - \ln\left(\min_{x^*\in\text{MinMaxStrats}_{opt}(A)} |\text{BR}(x^*)|\right)(1 + o_{\eta T}(1))\right),
\end{aligned}
$$

*where* $o_{\eta T}(1)$ *denotes a terms that decays to 0 as $\eta T \to \infty$, and $\text{BR}(x)$ is the set of best responses of a strategy $x$ of the optmizer as defined in 4.*

*Proof.* Fix some $x \in \Delta(\mathcal{A})$. Note that from Theorem 6 we have:

$$R^*_{cont}(\mathbf{0}, T, A, -A) \ge \frac{\ln(m)}{\eta} + \left(e_j^\top A^\top xT - \frac{1}{\eta}\ln\left(\sum_{i=1}^m e^{\eta(e_j^\top - e_i^\top)A^\top xT}\right)\right)$$

for any $j \in [m]$. Choose $j \in \text{BR}(x)$ (thus it minimizes $e_j^\top A^\top x$). Denote with $d_i(j, x) = (e_j - e_i)^\top A^\top x$. Note that for all $i \in [m]/\text{BR}(x)$ we have $d_i(j, x) = 0$ and for $i \notin \text{BR}(x)$ we have

$d_i(j, x) < 0$. We can rewrite this as:

$$R_{cont}^*(\mathbf{0}, T, A, -A)$$

$$\geq \frac{\ln(m)}{\eta} + \left( e_j^\top A^\top x T - \frac{1}{\eta} \ln\left( |\operatorname{BR}(x)| + \sum_{i \in [m]/\operatorname{BR}(x)} e^{\eta T \cdot d_i(j,x)} \right) \right)$$

$$= \frac{\ln(m)}{\eta} +$$

$$+ \left( e_j^\top A^\top x T - \frac{1}{\eta} \ln(|\operatorname{BR}(x)|) + \frac{1}{\eta} \ln\left( 1 + \frac{\sum_{i \in [m]/\operatorname{BR}(x)} e^{\eta T \cdot d_i(j,x)}}{|\operatorname{BR}(x)|} \right) \right) =$$

$$= \frac{\ln(m)}{\eta} +$$

$$+ \left( e_j^\top A^\top x T - \frac{1}{\eta} \ln(|\operatorname{BR}(x)|) \left( 1 + \frac{\ln\left( 1 + \frac{\sum_{i \in [m]/\operatorname{BR}(x)} e^{\eta T \cdot d_i(j,x)}}{|\operatorname{BR}(x)|} \right)}{\ln(|\operatorname{BR}(x)|)} \right) \right)$$

Note that as $\eta T \to \infty$, the term $\dfrac{\ln\left( 1 + \frac{\sum_{i \in [m]/\operatorname{BR}(x)} e^{\eta T \cdot d_i(j,x)}}{|\operatorname{BR}(x)|} \right)}{\ln(|\operatorname{BR}(x)|)}$ goes to 0, as $d_i(j, x) < 0$. Therefore this gives :

$$R_{cont}^*(\mathbf{0}, T, A, -A) \geq \frac{\ln(m)}{\eta} + \left( e_j^\top A^\top x T - \frac{1}{\eta} \ln(|\operatorname{BR}(x)|)(1 + o_{\eta T}(1)) \right)$$

Choosing $x \in \operatorname{MinMaxStrats}_{opt}(A)$ that minimizes $|\operatorname{BR}(x)|$ yields the desired result. $\qquad\square$

Below we provide an example of a game with multiple min-max strategies for the optimizer, however only one of them gives optimal rewards when used against MWU.

**Example 1.** *Consider the zero sum game where:*

$$A = $$

| | $b_1$ | $b_2$ | $b_3$ | $\ldots$ | $b_n$ | $b_{n+1}$ | $b_{n+2}$ | $b_{n+3}$ |
|---|---|---|---|---|---|---|---|---|
| $a_1$ | $n$ | 0 | 0 | $\ldots$ | 0 | $n$ | $n$ | 1 |
| $a_2$ | 0 | $n$ | 0 | $\ldots$ | 0 | $n$ | $n$ | 1 |
| $a_3$ | 0 | 0 | $n$ | $\ldots$ | 0 | $n$ | $n$ | 1 |
| $\vdots$ | $\vdots$ | $\vdots$ | $\vdots$ | $\vdots$ | $\ldots$ | $\vdots$ | $\vdots$ | $\vdots$ | $\vdots$ |
| $a_n$ | 0 | 0 | 0 | $\ldots$ | $n$ | $n$ | $n$ | 1 |
| $a_{n+1}$ | $n$ | $n$ | $n$ | $\ldots$ | $n$ | 2 | 0 | 1 |
| $a_{n+2}$ | $n$ | $n$ | $n$ | $\ldots$ | $n$ | 0 | 2 | 1 |

*Notice that the $\operatorname{Val}(A) = 1$, and there are multiple strategies that achieve this result, that might have a different number of best responses. For example $x_1 = (\frac{1}{n}, \frac{1}{n}, \ldots, \frac{1}{n}, 0, 0)$ has $n+1$ best responses; $\operatorname{BR}(x_1) = \{b_1, b_2, \ldots, b_{n+3}\}$. One of the strategies that achieves the value of the game with the least number of best responses is $x^* = (0, 0, \ldots, \frac{1}{2}, \frac{1}{2}, 0)$ that has only 1 best response, namely $\operatorname{BR}(x^*) = \{b_{n+3}\}$. For this game playing $x^*$ as $\eta T \to \infty$ yields $R_{cont}^*(\mathbf{0}, 0, A, -A) = T + \frac{\ln(n+3)}{\eta}$*

## B.2 Discrete Case

We can have the analogous definitions of optimal rewards for the optimizer for the discrete case. Given the sequence of actions of the optimizer given by $x : \{1, \ldots, T\} \to \Delta(\mathcal{A})$, we can define:

$$R_{disc}(x, h(0), T, A, B) = \sum_{t=1}^{T} \frac{\sum_{i=1}^{m} e^{\eta\left(h_i(0) + e_i^\top B^\top \sum_{s=1}^{t} x(s)\right)} \cdot e_i^\top A^\top x(t)}{\sum_{i=1}^{m} e^{\eta\left(h_i(0) + e_i^\top B^\top \sum_{s=1}^{t} x(s)\right)}} \tag{13}$$

Again, we are interested in finding the sequence of actions that maximize the rewards:

$$R_{disc}^*(h(0), T, A, B) = \max_{\substack{x(t) \\ 0 \le t \le T}} \sum_{t=1}^{T} \frac{\sum_{i=1}^{m} e^{\eta\left(h_i(0) + e_i^\top B^\top \sum_{s=1}^{t} x(s)\right)} \cdot e_i^\top A^\top x(t)}{\sum_{i=1}^{m} e^{\eta\left(h_i(0) + e_i^\top B^\top \sum_{s=1}^{t} x(s)\right)}} \tag{14}$$

We begin by proving that the rewards for the discrete game will always be more than the rewards for the analogous continuous game.

**Proposition 8.** *In the zero-sum discrete game, the optimizer can receive more utility compared to the continuous game. This can be done by playing discretely the same optimal strategy $x^*$ for the continuous game (as defined in Theorem 1). Consequently,*

$$R_{disc}^*(\mathbf{0}, T, A, -A) \ge R_{cont}^*(\mathbf{0}, T, A, -A) \tag{15}$$

*Proof.* From Theorem 1 we know that $R_{cont}^*(\mathbf{0}, 0, A, -A)$ is achieved by a constant strategy $x^*$ for the optimizer throughout the game. We will prove that using that same strategy $x^*$ will yield more reward for the discrete case than the continuous case, i.e.:

$$R_{disc}^*(\mathbf{0}, 0, A, -A) \ge \sum_{t=0}^{T-1} \frac{\sum_{i=1}^{m} e^{-\eta e_i^\top A^\top x^* t} \cdot e_i^\top A^\top x^*}{\sum_{i=1}^{m} e^{-\eta e_i^\top A^\top x^* t}}$$

$$\ge \int_{t=0}^{T} \frac{\sum_{i=1}^{m} e^{-\eta e_i^\top A^\top x^* t} \cdot e_i^\top A^\top x^*}{\sum_{i=1}^{m} e^{-\eta e_i^\top A^\top x^* t}} dt = R_{cont}^*(\mathbf{0}, 0, A, -A)$$

All we need for the above to be true is that the function $f(t) = \frac{\sum_{i=1}^{m} e^{-\eta e_i^\top A^\top x^* t} \cdot e_i^\top A^\top x^*}{\sum_{i=1}^{m} e^{-\eta e_i^\top A^\top x^* t}} = \frac{1}{\eta} \frac{\sum_{i=1}^{m} e^{-c_i t} \cdot c_i}{\sum_{i=1}^{m} e^{-c_i t}}$ is non-increasing. This is because $R_{disc}(\mathbf{0}, 0, A, -A) - R_{cont}(\mathbf{0}, 0, A, -A) = \sum_{t=0}^{T-1} f(t) - \int_0^T f(t) dt = \sum_{t=0}^{T-1} \left( f(t) - \int_t^{t+1} f(s) ds \right)$, so by proving $f$ is non-increasing we can prove $f(t) \ge \int_t^{t+1} f(s) ds$. Therefore, all we are left with is showing that $\frac{df(t)}{dt} < 0$ for all $t \ge 0$. Notice that:

$$f'(t) = -\frac{1}{\eta} \left( \frac{\sum_{i=1}^{m} e^{-c_i t} \cdot c_i^2}{\sum_{i=1}^{m} e^{-c_i t}} + \left( \frac{\sum_{i=1}^{m} e^{-c_i t} \cdot c_i}{\sum_{i=1}^{m} e^{-c_i t}} \right)^2 \right) < 0 \tag{16}$$

which concludes the proof. $\qquad\square$

We continue by proving that there are instances of discrete games, that achieve reward for the optimizer approximately $\frac{\eta T}{2}$ more than the analogous continuous game in Theorem 9. We also prove that this difference between rewards in continuous and discrete games cannot be greater than $\frac{\eta T}{2}$ in Theorem 10.

**Proposition 9.** *There is a zero sum game instance for which we have that*

$$R_{disc}^*(\mathbf{0}, T, A, -A) \ge R_{cont}^*(\mathbf{0}, T, A, -A) + \frac{\tanh(\eta) \cdot T}{2}$$

*The above is achieved by taking the optimal strategy $x^*$ for the continuous game (as defined in 1) and playing it over discrete time.*

*Proof.* Consider the matching pennies problem, where the utility matrices are:

$$A = \begin{array}{c|c|c} & b_1 & b_2 \\ \hline a_1 & 1 & -1 \\ \hline a_2 & -1 & 1 \end{array}, B = \begin{array}{c|c|c} & b_1 & b_2 \\ \hline a_1 & -1 & 1 \\ \hline a_2 & 1 & -1 \end{array}$$

Note that $\mathrm{Val}(A) = 0$, and $\mathrm{MinMaxStrats}_{opt}(A) = \{(\frac{1}{2}, \frac{1}{2})\}$, and $|\mathrm{BR}((\frac{1}{2}, \frac{1}{2}))| = 2$. This gives $R_{cont}^*(\mathbf{0}, T, A, -A) = 0$. On the other hand, consider the sequence of actions of the optimizer for the discrete game to be as follows:

$$x(t) = \begin{cases} a_2, & \text{if } t \text{ is odd} \\ a_1, & \text{otherwise} \end{cases}$$

We claim that at round $t$ where $t$ is odd, the learner will play $(\frac{1}{2}, \frac{1}{2})$, while when $t$ is even, the learner will play $(\frac{e^\eta}{e^\eta+e^{-\eta}}, \frac{e^{-\eta}}{e^\eta+e^{-\eta}})$. Indeed, at $t = 1, 3, 5, \ldots$, the optimizer will have played equal times the action $a_1$ and $a_2$, thus the rewards for actions $b_1$ and $b_2$ so far will be equal, leading the learner to play $(\frac{1}{2}, \frac{1}{2})$. When $t$ is even, action $a_2$ has been played one more time than $a_1$ is played. That means that at time $t$ for $b_1$ the historical reward is 1, while the historical reward for $b_2$ is $-1$. Thus the strategy for the learner for even $t$ will indeed be $(\frac{e^\eta}{e^\eta+e^{-\eta}}, \frac{e^{-\eta}}{e^\eta+e^{-\eta}})$. Therefore the optimizer's reward will be:

$$R_{disc}(x(t), \mathbf{0}, T, A, -A) = \frac{T}{2} \cdot [0 \quad 1] \begin{bmatrix} 1 & -1 \\ -1 & 1 \end{bmatrix} \begin{bmatrix} \frac{1}{2} \\ \frac{1}{2} \end{bmatrix} + \frac{T}{2} \cdot \begin{bmatrix} \frac{e^\eta}{e^\eta+e^{-\eta}} & \frac{e^{-\eta}}{e^\eta+e^{-\eta}} \end{bmatrix} \begin{bmatrix} 1 & -1 \\ -1 & 1 \end{bmatrix} \begin{bmatrix} 1 \\ 0 \end{bmatrix} = \tag{17}$$

$$= 0 + \frac{T}{2} \left( \frac{e^\eta}{e^\eta + e^{-\eta}} - \frac{e^{-\eta}}{e^\eta + e^{-\eta}} \right) = \frac{T}{2} \cdot \tanh(\eta) \approx \frac{\eta T}{2} \tag{18}$$

Combining the two gives the desired result. $\qquad\square$

**Proposition 10.** *In a zero-sum game where the entries of $A$ satisfy $A_{ij} \in [-1, 1]$, we have that the optimal utility the optimizer can achieve in the discrete game is not more than $\eta T/2$ compared to the continuous game:*

$$R^*_{disc}(\mathbf{0}, T, A, -A) - R^*_{cont}(\mathbf{0}, T, A, -A) \le \eta T/2 .$$

*Proof.* Suppose the optimal strategy for the discrete game is $x_{disc}$. We will construct a strategy $x_{cont}$ for the continuous game, that approximates the utility of the discrete game pretty well. The strategy is as follows; if $x_{disc}(t) = x_t \in \Delta(\mathcal{A})$, then $x_{cont}(s) = x_t, \forall a \in [t, t+1)$. Note that the historical rewards of the learner for $t = 0, 1, 2, \ldots, T-1$ are the same for both the continuous and the discrete game for these specific strategies. Suppose the history at time $t$ is $h(t)$, the strategy being played at time $t$ be $x$, and $u_i = e_i^\top A^\top x$. We have:

$$R^*_{disc}(h(t), 1, A, -A) = \frac{\sum_{i=1}^m e^{\eta h_i(t)} u_i}{\sum_{i=1}^m e^{\eta h_i(t)}}$$

We also have:

$$R^*_{cont}(h(t), 1, A, -A) = \int_0^1 \frac{\sum_{i=1}^m e^{\eta h_i(t) - \eta u_i t} \cdot u_i}{\sum_{i=1}^m e^{\eta h_i(t) - \eta u_i t}} dt \tag{19}$$

Denote by $f(t) = \frac{\sum_{i=1}^m e^{\eta h_i(t) - \eta u_i t} \cdot u_i}{\sum_{i=1}^m e^{\eta h_i(t) - \eta u_i t}}$. Then,

$$R^*_{disc}(h(t), 1, A, -A) = f(0),$$

whereas

$$R^*_{cont}(h(t), 1, A, -A) = \int_0^1 f(t) dt . \tag{20}$$

Note that:

$$f'(t) = \frac{-\eta \sum_{i=1}^m e^{\eta h_i(t) - \eta u_i t} \cdot u_i^2}{\sum_{i=1}^m e^{\eta h_i(t) - \eta u_i t}} + \frac{\eta \left( \sum_{i=1}^m e^{\eta h_i(t) - \eta u_i t} \cdot u_i \right)^2}{\left( \sum_{i=1}^m e^{\eta h_i(t) - \eta u_i t} \right)^2} .$$

Since $u_i$'s are bounded by in $[-1, 1]$, we get that $f'(t) \ge -\eta$. Hence $f(t) \ge f(0) - \eta t$, consequently,

$$R^*_{cont}(h(t), 1, A, -A) = \int_0^1 f(t) dt \ge \int_0^1 (f(0) - \eta t) dt = f(0) - \frac{\eta}{2} .$$

Therefore summing up for all $t = 0, 1, \ldots, T-1$:

$$R^*_{disc}(\mathbf{0}, T, A, -A) - R^*_{cont}(\mathbf{0}, T, A, -A) \le \eta T/2$$

$\qquad\square$

We conclude with the last proposition of this section which states that in games that satisfy the following Condition, we get that the optimizer can achieve reward at least $T \operatorname{Val}(A) + \Omega(\eta T)$ against a MWU learner.

**Condition 2.** *There exists a minmax strategy $x$ for the optimizer such that there exists two best responses for the learner, $b_{i_1}, b_{i_2} \in \mathrm{BR}(x)$, which do not identify on $\mathrm{support}(x)$. Namely, there exists an action $a_k \in \mathrm{support}(x)$ such that $a_k^\top A b_{i_1} \neq a_k^\top A b_{i_2}$.*

**Proposition 11.** *For any zero-sum game $A \in \mathbb{R}^{n \times n}$ that satisfies Condition 1, $\eta \in (0,1)$ and $T \in \mathbb{N}$, there exists an optimizer, such that against a learner that uses MWU with step size $\eta$, achieves a utility of at least*

$$T \,\mathrm{Val}(A) + C_A \eta T,$$

*where $C_A$ is a constant that depends only on the game matrix $A$.*

*Proof.* Let $x, b_{i_1}, b_{i_2}, a_k$ be the strategies and actions guaranteed from Condition 1. Since $b_{i_1}$ and $b_{i_2}$ are both best responses for $x$, $x^\top A e_{i_1} = x^\top A e_{i_2}$. By the Condition that there exists an action $a_k \in \mathrm{support}(x)$ such that $e_k^\top A e_{i_1} \neq e_k^\top A e_{i_2}$, it follows that there exists two strategies $x', x'' \in \Delta(A)$ such that $(x' + x'')/2 = x$, $x'^\top A e_{i_1} > x'^\top A e_{i_2}$ and $x''^\top A e_{i_1} < x''^\top A e_{i_2}$. We propose the following strategy for the learner: at any odd time $t$, play $x'$ and at any even time $t$, play $x''$. We will prove that in iterations $t$ and $t+1$ the reward of the optimizer is at least $2\,\mathrm{Val}(A) + C_A \eta$ by playing as described. Therefore, if we just sum up over all $t$, we will get the desired result. Now, fix some odd $t$, and we will lower bound the sum of utilities of the optimizer in iterations $t$ and $t+1$:

$$x(t)^\top A y(t) + x(t+1)^\top A y(t+1) = x'^\top A y(t) + x''^\top A y(t+1)$$

Denote $u_i = x'^\top A e_i$ and $v_i = x''^\top A e_i$, then

$$x'^\top A y(t) + x''^\top A y(t+1) = \sum_{i=1}^n u_i y_i(t) + \sum_{i=1}^n v_i y_i(t+1).$$

Notice that

$$y_i(t+1) = \frac{y_i(t) e^{-\eta x'^\top A b_i}}{\sum_{j=1}^n y_j(t) e^{-\eta x'^\top A b_j}} = \frac{y_i(t) e^{-\eta u_i}}{\sum_{j=1}^n y_j(t) e^{-\eta u_j}}.$$

Therefore

$$\sum_{i=1}^n u_i y_i(t) + \sum_{i=1}^n v_i y_i(t+1) = \sum_{i=1}^n u_i y_i(t) + \sum_{i=1}^n \frac{v_i(t) y_i(t) e^{-\eta u_i}}{\sum_{j=1}^n y_j(t) e^{-\eta u_j}}. \tag{21}$$

Notice that

$$\frac{u_i + v_i}{2} = \frac{(x' + x'')^\top}{2} A e_i = x^\top A e_i \geq \mathrm{Val}(A).$$

Hence, $v_i(t) \geq 2\,\mathrm{Val}(A) - u_i(t)$, hence the right hand side of Eq. (21) is at least

$$\sum_{i=1}^n y_i(t) u_i + 2\,\mathrm{Val}(A) - \sum_{i=1}^n \frac{u_i y_i(t) e^{-\eta u_i}}{\sum_{j=1}^n y_i(t) e^{-\eta u_i}} = 2\,\mathrm{Val}(A) + \sum_{i=1}^n y_i(t) u_i \left(1 - \frac{e^{-\eta u_i}}{\sum_{j=1}^n y_j(t) e^{-\eta u_j}}\right)$$

The right hand side equals $2\,\mathrm{Val}(A)$ if $\eta = 0$. We will differentiate it wrt $\eta$ to get

$$\frac{d}{d\eta}\left(-\frac{\sum_{i=1}^n y_i(t) u_i e^{-\eta u_i}}{\sum_{j=1}^n y_j(t) e^{-\eta u_j}}\right) = \frac{\sum_{i=1}^n y_i(t) u_i^2 e^{-\eta u_i}}{\sum_{j=1}^n y_j(t) e^{-\eta u_j}} - \left(\frac{\sum_{i=1}^n y_i(t) u_i e^{-\eta u_i}}{\sum_{j=1}^n y_j(t) e^{-\eta u_j}}\right)^2.$$

The right hand side equals the variance of a random variable, which we denote by $Z_\eta$, such that

$$\Pr[Z_\eta = u] = \sum_{i\,:\,u_i = u} \frac{y_i e^{-\eta y_i}}{\sum_{j=1}^n u_j e^{-\eta u_j}}.$$

We obtain that the sum of utilities for the optimzer in steps $t$ and $t+1$ is at least $2\,\mathrm{Val}(A) + \int_0^\eta \mathrm{Var}(Z_r) dr$. Recall that we assumed that $\eta \leq 1$. Hence, if we prove that there exists some constant $C_A$ such that $\mathrm{Var}(Z_r) \geq C_A$ for all $r \in [0,1]$, we will obtain that the sum of utilities in iterations $t$ and $t+1$ is at least $2\,\mathrm{Val}(A) + \eta C_A$. This will imply that the sum of utilities in iterations 1 through $T$ is at least $T\,\mathrm{Val}(A) + \eta T C_A / 2$, assuming that $T$ is even (if $T$ is odd then we can play a minmax strategy in the last round and get a similar bound). It remains to bound $\mathrm{Var}(Z_r)$. Recall that $b_{i_1}$ and $b_{i_2}$ are two best responses to $x$, and since $\sum_{i=1}^{t-1} x(i) = \frac{t}{2}x$, it holds from the definition of MWU that $y_{i_1}(t)$ and $y_{i_2}(t)$ are among the largest entries of $y(t)$. Consequently, $y_{i_1}(t), y_{i_2}(t) \geq 1/n$. This implies that there exists some $C > 0$ (that possibly depends on $A$ and $n$), such that for all $r \in (0,1)$, $\Pr[Z_r = u_{i_1}], \Pr[Z_r = u_{i_2}] \geq C$. Further, by Condition, $u_{i_1} = x'^\top A b_{i_1} \neq x'^\top A b_{i_2} = u_{i_2}$. Consequently, there exists some $C_A > 0$ such that $\mathrm{Var}(Z_r) \geq C_A$ for all $r \in [0,1]$. This is what we wanted to prove. $\qquad\square$

## C  Computational lower bound

In this section, we present the full omitted proof of Theorem 4.

*Proof of theorem 4.* We'll show a polynomial time reduction from OCDP to the hamiltonian cycle problem. Let $n = |V|$ and $m = |E|$. Given an instance $\pi_{Ham} = G(V, E)$, where $V = \{v_1, v_2, \ldots, v_n\}$ and $E = \{e_1, \ldots, e_m\}$ of the Hamiltonian cycle problem, we create an instance of the OCDP problem as follows: the optimizer has $m$ actions, denoted by $\mathcal{A} = \{a_1, \ldots, a_m\}$ and the learner has $2n$ actions, denoted by $\mathcal{B} = \{b_1, \ldots, b_{|V|}, b_{\text{in}_1}, \ldots, b_{\text{in}_{|V|}}\}$. The matrices $A$ and $B$, which correspond to the optimizer and learner's utility matrices respectively are defined as follows:

$$A[a_i, b_j] = \begin{cases} 1, & \text{if there exists } u \in V \text{ such that } e_i = (v_j, u), \text{ i.e. } e_i \text{ is outgoing edge of node } v_j \\ 0, & \text{otherwise} \end{cases}$$

$$A[a_i, b_{\text{in}_j}] = 0$$

$$B[a_i, b_j] = \begin{cases} -0.1 & \text{if } j = 1 \text{ and } \exists u \in V \text{ s.t. } e_i = (v_1, u), \text{ i.e. } e_i \text{ is an outgoing edge of node } v_1 \\ -4, & \text{if } j \neq 1 \text{ and } \exists u \in V \text{ s.t. } e_i = (v_j, u), \text{ i.e. } e_i \text{ is an outgoing edge of node } v_j \\ 1, & \text{if } \exists u \in V \text{ such that } e_i = (u, v_j), \text{ i.e. } e_i \text{ is an incoming edge of node } v_j \\ 0, & \text{o.w.} \end{cases}$$

$$B[a_i, b_{\text{in}_j}] = \begin{cases} 0.85, & \text{if } \exists u \in V \text{ such that } e_i = (v_j, u), \text{ i.e. } e_i \text{ is an outgoing edge of node } v_j \\ 0, & \text{o.w.} \end{cases}$$

Finally, we set $k = T = n + 1$, constructing the instance $\pi_{OCDP} = (A, B, m, 2n, k, T)$. We conclude the reduction by proving that $\pi_{Ham}$ is a 'Yes' instance if and only if $\pi_{OCDP}$ is a 'Yes' instance.

1. $\pi_{\textbf{Ham}} \implies \pi_{\textbf{OCDP}}$ : Suppose we have a Hamiltonian cycle that visits vertices $v_1 = v_{u_1} \to v_{u_2} \to \cdots \to v_{u_n} \to v_{u_{n+1}} = v_{u_1} = v_1$. Suppose that edge $e_{p_i}$ connects $v_{u_i}$ to $v_{u_{i+1}}$. We will prove that the sequence of actions $a_{p_1}, a_{p_2}, \ldots, a_{p_{n-1}}, a_{p_n}, a_{p_{n+1}} = a_{p_1}$ achieve reward exactly $n+1$ for the optimizer. To prove that, it is enough to argue that if the optimizer plays as we described above, the learner will respond with $b_{u_1}, b_{u_2}, \ldots, b_{u_n}, b_{u_{n+1}} = b_{u_1}$. Indeed, notice that $A[a_{p_i}, b_{u_i}] = 1, \forall i \in [n+1]$, since $e_{p_i}$ is an outgoing edge of $v_{u_i}$, therefore the optimizer will be getting reward 1 every round. Now, to prove that the learner best responds as described, let us look at the historical rewards of the learner. Suppose the rewards for each of the $m = 2 \cdot |V|$ actions of the learner for rounds $1, 2, \ldots, t-1$ is denoted by $h(t) = (h_1(t), \ldots, h_n(t), h_{\text{in}_1}(t), \ldots, h_{\text{in}_n}(t)) \in \mathbb{R}^{2n}$. We will prove inductively that after $t$ rounds we will have $h(t) = H(t)$, where $H(t)$ is defined as:

$$H_1(t) = \begin{cases} 0, & \text{if } t = 1 \\ -0.1, & \text{if } 2 \leq t \leq n \\ 0.9, & \text{if } t = n + 1 \end{cases}$$

$$H_{u_i}(t) = \begin{cases} 0, & \text{if } t < i \\ 1, & \text{if } t = i, i > 1 \\ -3, & \text{if } t > i \end{cases}$$

$$H_{\text{in}_{u_i}}(t) = \begin{cases} 0, & \text{if } t \leq i \\ 0.85, & \text{if } t > i \end{cases}, i = 1, 2, \ldots, n$$

Notice that the cases $t = 1, 2$ are trivial; at $t = 1$ everything is equal to 0 since the learner has accumulated no reward yet. At time $t = 2$ we only update $h_{\text{in}_1} \to 0.85, h_1 \to -0.1, h_{u_2} \to 1$, thus giving us $h(2) = H(2)$. Suppose that after some $t$ rounds, where $2 \leq t \leq n$ rounds we have $h(t) = H(t)$. After $a_{p_t}$ is played by the optimizer, the learner has to update $h_{\text{in}_{u_t}} \to 0.85, h_{u_t} \to 1 - 4 = -3$ and if $t < n$ we update $h_{u_{t+1}} \to 1$ otherwise if $t = n$ we update $h_1 \to -0.1 + 1 = 0.9$. Either way, we get $h(t+1) = H(t+1)$. To complete the proof, note that for each $t$, given that the of the learner history is $H(t)$, best response for the learner is $b_{u_t}$, thus the sequence of actions $a_{p_1}, a_{p_2}, \ldots, a_{p_n}, a_{p_{n+1}}$ achieves reward $n + 1$ for the optimizer.

2. $\pi_{\textbf{OCDP}} \implies \pi_{\textbf{Ham}}$ : Suppose there is a sequence of actions played by the optimizer that receive reward $n + 1$ in $n + 1$ rounds, with actions $a_{p_1}, a_{p_2}, \ldots, a_{p_{n+1}}$. We will prove that $a_{p_1}, \ldots a_{p_n}$ correspond to the edges of a Hamiltonian Cycle starting from node $v_1 = 1$, i.e. edges $e_{p_1}, \ldots, e_{p_n}$ make a Hamiltonian Cycle, by connecting nodes $v_1 = v_{u_1}, v_{u_2}, \ldots, v_{u_n}$. We again denote the rewards of the learner for rounds $1, 2, \ldots, t-1$ with $h(t) = (h_1(t), \ldots, h_n(t), h_{\text{in}_1}(t), \ldots, h_{\text{in}_n}(t)) \in \mathbb{R}^{2n}$.

- **Action at $t = 1$:** Start by observing that the learner will play $b_1$ at $t = 1$ as it is the tie breaking rule, so the optimizer needs to play an action that achieves reward 1 against that. Note that the way we constructed the utility matrix of the optimizer, the only actions that achieve reward against $v_i$ correspond to outgoing edges from vertex $v_i$. So at $t = 1$, the optimizer is forced to play an action that corresponds to an outgoing edge of $v_1 = 1$. Suppose the first action of the optimizer is $a_{p_1}$, corresponding to the edge $e_{p_1} = (v_1, v_{u_2})$.

- **Action at $t = 2$:** The rewards of the optimizer will be updated as follows, after the optimizer plays $a_{p_1}$ and the learner plays $b_1$:

$$H_i(t) = \begin{cases} -0.1, & \text{if } i = 1 \\ 1, & \text{if } i = u_2 \\ 0, & \text{o.w.} \end{cases}$$

$$H_{\text{in}_i}(t) = \begin{cases} 0.85, & \text{if } i = 1 \\ 0, & \text{o.w.} \end{cases}$$

We note that now new best response of the learner will be $b_{u_2}$. Thus, the only way for the optimizer to obtain reward 1 for the second round is to play an action corresponding to an outgoing edge of node $v_{u_2}$. Suppose the action the optimizer plays is $b_{p_2}$, corresponding to an edge $e_{p_2} = (v_{u_2}, v_{u_3})$.

- **Actions at $2 < t < n$:** We will prove inductively that the first $t \leq n - 1$ actions $a_{p_1}, \ldots, a_{p_t}$ correspond to edges of the form $e_{p_1} = (v_{u_1}, v_{u_2}), e_{p_2} = (v_{u_2}, v_{u_3}), \ldots, e_{p_t} = (v_{u_t}, v_{u_{t+1}})$ where $u_i \neq u_j$ for $1 \leq i < j \leq n$, i.e. edges that define a simple path in the graph, starting from node $v_1$. By playing actions in such a manner, the historical rewards of the learner will be of the form:

$$H_1(t) = \begin{cases} 0, & \text{if } t = 1 \\ -0.1, & \text{if } 2 \leq t \leq n \end{cases}$$

$$H_{u_i}(t) = \begin{cases} 0, & \text{if } t < i \\ 1, & \text{if } t = i\,, i > 1 \\ -3, & \text{if } t > i \end{cases}$$

$$H_{\text{in}_{u_i}}(t) = \begin{cases} 0, & \text{if } t \leq i \\ 0.85, & \text{if } t > i \end{cases}, i = 1, 2, \ldots, n$$

We proved that the first $t$ actions correspond to edges of a simple path for $t \leq 2$. We assume for the inductive step that it is true for the first $t < n-1$ actions $a_{p_1}, \ldots, a_{p_t}$ of the optimizer. We will prove that the next action $a_{p_{t+1}}$ by the optimizer corresponds to an edge $e_{p_{t+1}}$ that extends this simple path. Since the first $t$ actions are corresponding to a simple path, at round $t + 1$ the learner's historical rewards are going to be equal to $H(t + 1)$. Note that the best response for the learner is going to be $b_{u_{t+1}}$. In order for the optimizer to gain reward 1 the $t+1$'th round, they have to be playing an action $a_{p_{t+1}}$ for which $e_{p_{t+1}}$ is an outgoing edge of the node $v_{u_{t+1}}$. Suppose the optimizer chooses to play $a_{p_{t+1}}$, where $e_{p_{t+1}} = (v_{u_{t+1}}, v_{u_{t+2}})$, and $u_{t+2} = u_l$ for some $1 \leq l \leq t + 1$. At round $t + 2$ the optimizer will still have to play an action that corresponds to an outgoing edge of $v_{u_l}$. However, after that, we will have $h_{\text{in}_{u_l}}(t + 3) = 1.70$, which will make $b_{u_l}$ the best response. However, there is no action the optimizer can play against $b_{u_l}$ in which they receive reward 1, making it impossible for the optimizer to obtain $n + 1$ reward in $n + 1$ rounds. Thus the optimizer needs to play $e_{p_{t+1}} = (v_{u_{t+1}}, v_{u_{t+2}})$, and $u_{t+2} \neq u_l, l \in [t + 1]$, completing the inductive step. Since we also proved that the first 2 actions are forming a simple path starting from $v_1$, using the inductive step, we conclude that the first $n - 1$ actions have to correspond to a simple path in the graph starting from $v_1$.

- **Actions at $t = n, n+1$:** Suppose the last two actions of the optimizer are $a_{p_n}, a_{p_{n+1}}$. We want to prove that $e_{p_n} = (v_{u_n}, v_1), e_{p_{n+1}} = (v_1, u)$, where $u$ is some node connected to $v_1$. We know that the history of the learner at time $t = n$, as proven in the previous bullet point, is as follows:

$$
h_{u_i}(n) = \begin{cases} -0.1, & \text{if } i = 1 \\ 1, & \text{if } i = n, i > 1 \\ -3, & \text{o.w.} \end{cases}
$$

$$
h_{\text{in}_{u_i}}(n) = \begin{cases} 0, & \text{if } i = n \\ 0.85, & \text{o.w.} \end{cases}, i = 1, 2, \ldots, n
$$

Note that the best response for the learner at time $t = n$ will be $b_{u_n}$. Thus the optimizer will have to play an action $a_{p_n}$ corresponding to an outgoing edge of $v_{u_n}$. Suppose $e_{p_n} = (v_{u_n}, x)$. Assume $x \neq 1$. Then the history is updated to $h_{u_n} \to -3, h_x \to -2, h_{\text{in}_{u_n}} \to 0.85$. Note that the best response is any action $b_{u_{\text{in}_i}}$ for which the optimizer cannot get reward 1 from. Thus if $x \neq 1$ it is impossible to get reward 1 every round. On the other hand, if $x = 1$, then the best response at time $n + 1$ will be $b_1$ since the $h_1(n + 1) = -0.1 + 1 = 0.9 > 0.85$, and thus playing any action corresponding to an outgoing edge of $v_1$ will obtain reward 1 for the learner at that round.

Lastly, notice that the problem statement requests the entries of the weight matrix $B$ to be in $[0, 1]$. While our construction have weights in $[-4, 4]$, scaling and shifting all the entries by the same amount does not affect the identity of the best response. □

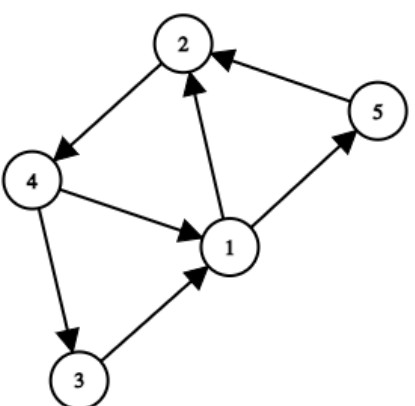

Figure 1: Graph $G$

We will show an example of how we can make the reduction from Hamiltonian Cycle to OCDP. Imagine we have an instance of Hamiltonian Cycle that corresponds to the graph shown in 1, and we want to reduce to OCDP. Since we have $|V| = 5$ vertices and $|E| = 7$ edges, we instantiate the following instance of OCDP: $(A, B, n = |E|, m = 2|V|, k = |V| + 1, T = |V| + 1)$, where the actions sets are $\mathcal{A} = \{e_1 = (1, 5), e_2 = (5, 2), e_3 = (1, 2), e_4 = (2, 4), e_5 = (4, 1), e_6 = (4, 3), e_7 = (3, 1)\}$ $\mathcal{B} = \{v_1, v_2, v_3, v_4, v_5, v_{in_1}, v_{in_2}, v_{in_3}, v_{in_4}, v_{in_5}\}$, and the utility matrices $A$ and $B$ as shown in the tables below.

|       | $v_1$ | $v_2$ | $v_3$ | $v_4$ | $v_5$ | $v_{in_1}$ | $v_{in_2}$ | $v_{in_3}$ | $v_{in_4}$ | $v_{in_5}$ |
|-------|-------|-------|-------|-------|-------|-----------|-----------|-----------|-----------|-----------|
| $e_1$ | 1     | 0     | 0     | 0     | 0     | 0         | 0         | 0         | 0         | 0         |
| $e_2$ | 0     | 0     | 0     | 0     | 1     | 0         | 0         | 0         | 0         | 0         |
| $e_3$ | 1     | 0     | 0     | 0     | 0     | 0         | 0         | 0         | 0         | 0         |
| $e_4$ | 0     | 1     | 0     | 0     | 0     | 0         | 0         | 0         | 0         | 0         |
| $e_5$ | 0     | 0     | 0     | 1     | 0     | 0         | 0         | 0         | 0         | 0         |
| $e_6$ | 0     | 0     | 0     | 1     | 0     | 0         | 0         | 0         | 0         | 0         |
| $e_7$ | 0     | 0     | 1     | 0     | 0     | 0         | 0         | 0         | 0         | 0         |

A =

|       | $v_1$ | $v_2$ | $v_3$ | $v_4$ | $v_5$ | $v_{in_1}$ | $v_{in_2}$ | $v_{in_3}$ | $v_{in_4}$ | $v_{in_5}$ |
|-------|-------|-------|-------|-------|-------|-----------|-----------|-----------|-----------|-----------|
| $e_1$ | -1    | 0     | 0     | 0     | 1     | 0.85      | 0         | 0         | 0         | 0         |
| $e_2$ | 0     | 1     | 0     | 0     | -4    | 0         | 0         | 0         | 0         | 0.85      |
| $e_3$ | -1    | 1     | 0     | 0     | 0     | 0.85      | 0         | 0         | 0         | 0         |
| $e_4$ | 0     | -4    | 0     | 1     | 0     | 0         | 0.85      | 0         | 0         | 0         |
| $e_5$ | 1     | 0     | 0     | -4    | 0     | 0         | 0         | 0         | 0.85      | 0         |
| $e_6$ | 0     | 0     | 1     | -4    | 0     | 0         | 0         | 0         | 0.85      | 0         |
| $e_7$ | 1     | 0     | -4    | 0     | 0     | 0         | 0         | 0.85      | 0         | 0         |

B =

Notice that the only way for the optimizer to achieve reward $|V| + 6 = 5 + 1 = 6$ is if he plays the following actions:

$$e_1, e_2, e_4, e_6, e_7, e_1$$

and the learner's actions are the following:

$$v_1, v_5, v_2, v_4, v_3, v_1$$

The following table shows the rewards history of the learner during this game:

|         | $v_1$ | $v_2$ | $v_3$ | $v_4$ | $v_5$ | $v_{in_1}$ | $v_{in_2}$ | $v_{in_3}$ | $v_{in_4}$ | $v_{in_5}$ |
|---------|-------|-------|-------|-------|-------|-----------|-----------|-----------|-----------|-----------|
| $t = 0$ | 0     | 0     | 0     | 0     | 0     | 0         | 0         | 0         | 0         | 0         |
| $t = 1$ | -0.1  | 0     | 0     | 0     | 1     | 0.85      | 0         | 0         | 0         | 0         |
| $t = 2$ | -0.1  | 1     | 0     | 0     | -3    | 0.85      | 0         | 0         | 0         | 0.85      |
| $t = 3$ | -0.1  | -3    | 0     | 1     | -3    | 0.85      | 0.85      | 0         | 0         | 0.85      |
| $t = 4$ | -0.1  | -3    | 1     | -3    | -3    | 0.85      | 0.85      | 0         | 0.85      | 0.85      |
| $t = 5$ | -0.1  | -3    | 1     | -3    | -3    | 0.85      | 0.85      | 0         | 0.85      | 0.85      |
| $t = 5$ | 0.9   | -3    | -3    | -3    | -3    | 0.85      | 0.85      | 0.85      | 0.85      | 0.85      |
| $t = 6$ | 0.8   | -3    | -3    | -3    | -3    | 1.70      | 0.85      | 0.85      | 0.85      | 0.85      |

r(t) =

