# OpenReview forum: "Maximizing utility in multi-agent environments by anticipating the behavior of other learners"
_NeurIPS.cc/2024/Conference — NeurIPS 2024 poster_

### Official Review · Reviewer_PJLW · 2024-07-07

**Soundness:** 4
**Presentation:** 3
**Contribution:** 3
**Rating:** 6
**Confidence:** 2

**Summary:**

- The authors consider the problem of optimally exploiting a learning agent in zero-sum games and general-sum games.
- For zero-sum games, they look at a specific continuous time learner, replicator dynamics, and at an analogous discrete time learner, multiplicative weights updates. They provide an explicit expression for optimal utility of the exploiter agent in the continuous case, as well as a construction for a polynomial-time algorithm that achieves that utility.
- In the discrete case, they show that the discretized optimal strategy from the continuous case achieves at least as much utility. Moreover, they show that this utility can be exceeded, and they bound this difference between the continuous and discrete cases from below and from above.
- In the context of general-sum games, the paper shows that determining the optimal utility of the exploiter against a Best Response Algorithm (in discrete time) is NP-hard.
- The paper concludes by discussing open problems, in particular whether there are some learners and classes of general-sum games where one can say more about the optimal utilities for polynomial-time optimizers.

**Strengths:**

- This paper makes a significant contribution by analyzing the general setting in which an optimizer is trying to exploit a mean-based learner. In my view, both the impossibility result in the general case as well as the characterization of the zero-sum case are conceptually interesting.
- The mathematical analysis appears sound and rigorous. The paper discusses an interesting connection to optimal control. (Though I have not checked the proofs in the appendix and this is not my main area of expertise).
- I found the paper was well written and well motivated.
- I appreciated the discussion of societal impact. I agree with the authors that this work is relevant since it is important to study how exploitable commonly used learning algorithms are.

**Weaknesses:**

- The paper doesn't include any empirical simulations. It would have been interesting to see some example numerical computations of the optimal strategy in a game and resulting simulated utilities. How much better does the optimal strategy do compared to simple baselines, in some canonical games?
- Focusing on MWU and related algorithms makes sense to make the analysis tractable, but it also limits the applicability of the results. Moreover, it would be especially interesting to have an impossibility result for a no regret algorithm.
- While the computational hardness result is interesting, it does not appear very surprising given typical hardness results in game theory.
- While I found that the authors generally explained their approach well, I thought it might have been interesting to give some more explanation, potentially via a concrete example (in the zero-sum case). If this takes up too much space, one could include an example in the appendix.

**Questions:**

- I wonder whether it would be possible to give a description of the constant optimal strategy in the continuous-time case, in words. How does it relate to the minimax strategy? I assume that even as $T\rightarrow \infty$, learners play suboptimally if the temperature $\eta<\infty$, and so the optimizer would deviate from the minimax strategy somewhat to exploit this. However, is it right that for $\eta\rightarrow \infty$, the optimizer strategy would eventually converge to the minimax strategy?
- Line 106: Should it say "$\eta\rightarrow \infty$ as $T\rightarrow \infty$"?
- This work relates to empirical ML work in which learners are gradient based and exploitative strategies are found via meta optimization (see e.g. model-free opponent shaping, https://arxiv.org/abs/2205.01447, and follow up work)
- Could you say under what conditions the constant $C_A$ in Proposition 4 is small? Does this depend on the inequality in Assumption 1? (i.e., if the two best responses lead to very different utilities, this leads to more possibility of exploitation?)
- Typo on line 350: "The details the force the learner"
- In the related work section, it would be nice to better contrast the discussed related work (especially on no regret learners) to the work in this paper. E.g., MWU is also a no regret learner, but best response isn't? How exactly does this paper extend prior work on exploiting these different learners? E.g., by analyzing computational complexity, or by providing tighter bounds, etc.?

**Limitations:**

The authors adequately address limitations.

---

> ### Author Rebuttal · Authors · 2024-08-07
>
> We thank the reviewer for the useful comments. We answer the main questions below, and move a few minor questions to an official comment, for space considerations.
>
> ``It would have been interesting to see some example numerical computations of the optimal strategy in a game and resulting simulated utilities. How much better does the optimal strategy do compared to simple baselines, in some canonical games?''
>
> We are happy to include examples of games and the optimal rewards and strategies of the optimizer for different time horizons $T$ in the new revision of the paper.
>
> ``Focusing on MWU and related algorithms makes sense to make the analysis tractable, but it also limits the applicability of the results. Moreover, it would be especially interesting to have an impossibility result for a no regret algorithm.''
>
> Even understanding what happens against specific types of algorithms such as MWU is an elusive open problem (look at paragraph 2 of page 3 of Is learning in games good for the learners), but also understanding what the optimal strategies are against these algorithms will be helpful for the analysis against other no-regret learners. You are right, an impossibility result for no-regret learning algorithms would be extremely interesting, and we leave it open for future work.
>
> ``While the computational hardness result is interesting, it does not appear very surprising given typical hardness results in game theory.''
>
> A lot of results in game theory are impossibility results (such as for example computation of several types of equilibria in different games). However, our impossibility result is the first of its kind, as far as we know. That is, our result is the first that states that computing optimal strategies against learners with specific dynamics can be computationally hard, and none of the previous work has talked about any computational issues thus far.
>
> ``I wonder whether it would be possible to give a description of the constant optimal strategy in the continuous-time case, in words. How does it relate to the minimax strategy?''
>
> As $\eta T \to \infty$ then the optimal constant strategy will approach a min-max strategy. However, for finite or small $T$ the optimal strategy might be different, depending on the game.
>
> ``I assume that even as $T\to \infty$, learners play suboptimally if the temperature $\eta \to \infty$, and so the optimizer would deviate from the minimax strategy somewhat to exploit this. However, is it right that for $\eta \to \infty$, the optimizer strategy would eventually converge to the minimax strategy?''
>
> If $\eta \to \infty$ then the learner is essentially employing fictitious play (best responding to the history). When $\eta \to \infty$, the optimizer is able to take advantage of that by significantly deviating from the min-max strategy. This can also be seen in the way we analyze the computational lower bound and specifically you can view it in the example we provide in the last pages of the paper (p22-23). There, optimizer constantly changes the actions they are playing. The phenomenon where the optimizer has to constantly switch actions can also be seen in the matching pennies example that we also discuss in the paper (Proposition 9). It certainly is the case for matching pennies that the average-time strategy for the optimizer in the scenario where $\eta$ is large will converge to the min-max strategy, however, the per-round strategy is not. It is an interesting question we have not tackled whether the average-time strategy always converges to the minmax strategy.
>
> ``Could you say under what conditions the constant $C_A$ in Proposition 4 is small? Does this depend on the inequality in Assumption 1? (i.e., if the two best responses lead to very different utilities, this leads to more possibility of exploitation?)''
>
> Yes, this depends on the inequality in Assumption 1. If the assumption is close to being unsatisfied, then the constant is small. If the assumption is satisfied with a large gap, then the constant is larger, which implies a better lower bound on the utility gained by the optimizer.
>
> ``In the related work section, it would be nice to better contrast the discussed related work (especially on no regret learners) to the work in this paper. E.g., MWU is also a no regret learner, but best response isn't? How exactly does this paper extend prior work on exploiting these different learners? E.g., by analyzing computational complexity, or by providing tighter bounds, etc.?''
>
> Thanks for the question. Previous work has introduced the problem of strategizing against no-regret learners, and specifically mean based learners. It has also been shown that in zero sum games against these mean based learners, the best rewards the optimizer can achieve are $T \cdot Val(A) + o(T)$, while in general sum games, it has been shown that the optimizer can get significantly more utility than the one shot game value (which for general sum games is called the Stackelberg value), however no efficient algorithm on how to achieve this value has been found. In our work, we show for zero-sum games what is the exact optimal rewards for the optimizer for Replicator dynamics - one of the most widely used mean base learners. We also show a first computational lower bound for computing the optimal strategy against a learner that is best responding, i.e. MWU with infinite step size. We believe that ultimately the problem of computing an optimal strategy against mean based learners in general sum games is computationally hard, and we give a first result with our lower bound towards that direction. We will add these details in the related work section.

---

> ### Author Response · Authors · 2024-08-07
> **Additional responses to questions of the reviewer**
>
> ``While I found that the authors generally explained their approach well, I thought it might have been interesting to give some more explanation, potentially via a concrete example (in the zero-sum case). If this takes up too much space, one could include an example in the appendix.''
>
> We have a few examples (Example 1 in the appendix and the matching pennies game in Proposition 9), and we are happy to include more examples of zero sum games that show what the rewards of the optimizer are.
>
>
> ``Line 106: Should it say $\eta \to \infty$ as $T \to \infty$''
>
> No this is correct as is. Usually the step size is set to be small i.e. $\eta = \frac{1}{\sqrt{T}}$ so increasing $T$ makes $\eta$ go to $0$.
>
> ``Typo on line 350: "The details the force the learner"''
>
> Thanks for point that out. It should be "The details of how to force the learner ..."

---

> > ### Comment · Reviewer_PJLW · 2024-08-11
> >
> > Thanks a lot for your detailed clarifications.
> >
> > (Regarding the $\eta\rightarrow\infty$ point: thank you, I was confused about this. I think I understand now that $\eta T$ should go to infinity, but $\eta$ should go to zero, to enable learning but to prevent an oscillating/exploitable policy.)

---

### Official Review · Reviewer_jNhL · 2024-07-11

**Soundness:** 3
**Presentation:** 1
**Contribution:** 2
**Rating:** 3
**Confidence:** 4

**Summary:**

The authors present a model where an optimizer plays with a learning agent and aims to extract better rewards from the sequential decision-making game by anticipating what the learner will do selecting a strategy that outperforms the value of a game. They study two settings: a zero-sum game, where they show a polynomial time algorithm that can extract an advantage for the optimizer, and a general-sum game, where they show the problem is NP-hard by reduction from the Hamiltonian cycle problem. They show that for a learner using Replicator Dynamics, the optimizer can get an advantage related to the time horizon by using a constant strategy.

**Strengths:**

The paper tackles the important problem of using information about other agents to maximize utility in two-player games. The authors present various theorems and proofs to show how strategies for the optimizer can be devised to gain an advantage. Further, they also show a novel hardness proof for the general-sum game by reduction from the Hamiltonian cycle problem, when playing against a mean-based learner.

**Weaknesses:**

My primary concern with this paper is in the organization and clarity of presentation. The authors do not contextualize and clearly state their results, or talk about the importance of their results. Further, a large portion of the theoretical information is included in the introduction, with various theorems being re-stated later on. Then, there is an awkward interlude with the related work section before continuing with the remaining theoretical analysis. This makes the paper very hard to follow, without any appropriate structure to guide the reader. I would expect a much shorter introduction, which introduces the problem setting and introduction without going into theoretical detail and summarizes the paper's contributions, following which related work or background for the problem is provided. In general, main Theorems and Propositions should not appear in the Introduction.

Some important details which could be highlighted are missing from the body. For example, the authors don't clearly state the benefit that can be achieved in a zero-sum game by the optimizer, except for the informal statement of proposition 1. This seems to me one of the main contributions of the paper, other than the hardness proof.

Theorems and propositions are referred to by multiple names and repeated. If possible, authors should use the same numbers for the same theorem or proposition.

The authors also focus only on the MWU and Replicator Dynamics models for the learner. However, there is no justification for why this model is chosen as opposed to any other kind of learner. The setting described is general, however all proofs rely on the learner using this model. Some discussion on why MWU and replicator dynamics are chosen will add to the concreteness of the paper.

Some minor/specific comments below:
It would be helpful if n and m are described explicitly as the cardinality of the action spaces of the two players. As it is right now, it is easy to miss when they are first introduced, and later equations (e.g. Eq. 1) rely on the readers knowing what they stand for.

What is \Delta, as used in the definition of the value for zero-sum games? It seems to be used as a function, but it isn't clear from the context what it is supposed to be. This same notation is used in line 163, which suggests \Delta(A) might be a set?

Line 164: i1, i2 are not described in Proposition 4.

The replicator dynamics is defined as a continuous time analogue of the MWU algorithm, but the MWU algorithm itself is not described until later in the paper. This can be confusing for readers, if MWU is not being used, then it should not be introduced earlier. If Replicator Dynamics needs to be described as a generalization or adaptation of MWU, MWU needs to be described first.

**Questions:**

1. Can you justify why MWU and Replicator Dynamics are chosen as the model for the learner?
2. Can you clearly state what the novel contribution of the paper is? My current understanding is that the hardness proof is novel, and the polynomial time algorithm for Zero-sum games is novel, and that is all.
3. Can you clarify the undefined notation, e.g. \Delta(A)?
4. Why is so much exposition included in the Introduction? Is there a reason for this?

**Limitations:**

The authors don't specifically have a limitations section addressing the limits of the work, though they do propose future work building on the current results. The authors should address why only one model of the learner is considered, as that can be seen as a limitation.

---

> ### Author Rebuttal · Authors · 2024-08-07
>
> We thank the reviewer for their feedback. We answer below the main concerns, and we answer additional questions in an official comment, due to the space limitations.
>
> ``My primary concern with this paper is in the organization and clarity of presentation...'':
>
> We thank the author for their comment about the presentation. We note that in theory papers it is common to have an informal version of the theorems in the introduction, followed by a longer version in the technical sections. Yet, we will change the format in accordance to the reviewer's comments, and we agree with the reviewer that such a change can benefit the paper. Specifically, after the meta question, we will summarize in bullets the main contributions of the paper. Related work will follow. Then, we will write the preliminaries, and afterwards the results, in two sections: (1) optimizing against MWU and (2) a lower bound for optimizing against Best-Response. These will include results that currently appear both in the introduction and in sections 2 and 3.
>
> ``Some important details which could be highlighted are missing from the body. For example, the authors don't clearly state the benefit that can be achieved in a zero-sum game by the optimizer, except for the informal statement of proposition 1. This seems to me one of the main contributions of the paper, other than the hardness proof.''
>
> We state the benefit through theorem 1 and propositions 1 to 4 in the introduction. First of all, theorem 1 states that the optimal rewards for the optimizer can be achieved with a strategy that is constant over time (i.e. $x(t) = x^*, 0\leq t \leq T$). In proposition 1 we state exactly the range of possible rewards the optimizer can achieve and what happens when $\eta T \to \infty$. In propositions 2-4 we explain what happens in the games with discrete time dynamics. First of all, the discrete time analogue of a continuous time game will always give more rewards for the optimizer (Proposition 2). This 'gap' in the rewards cannot be more than $\eta T/2$ and there exists a game that approximately achieves this gap, namely matching pennies (Proposition 3). Lastly, we show that up to constant factors, most games also exhibit this reward gap between the continuous and discrete game analogues (proposition 4).
>
> ``Can you justify why MWU and Replicator Dynamics are chosen as the model for the learner?''
>
> We chose the MWU algorithm for the learner for several reasons. First of all, it is one of the most well-studied no-regret algorithms due to its optimal regret but also as its continuous time analogue, the replicator dynamics, is extremely well studied in the literature. MWU is also an algorithm that is widely used in online learning settings, therefore knowing how robust or manipulable it can be against strategic players in online settings is an interesting question. We also believe that our view of the problem as a continuous time problem, provides a new perspective on how to view the problem of strategizing against mean based learners (as introduced by [1]).
>
> ``Can you clearly state what the novel contribution of the paper is? My current understanding is that the hardness proof is novel, and the polynomial time algorithm for Zero-sum games is novel, and that is all.''
>
> All of the results in our paper are novel, and are the following:
>
> (1) The first set of results concerns zero-sum games where the learner is employing Replicator dynamics (for the coninuous case) and MWU (for the discrete case), as summarized in (1a) and (1b) below:
>
> (1a) We show what is the exact best strategy for the optimizer against a replicator dynamics learner in the continuous time setting by presenting a closed form solution (Theorem 1). We show that this strategy can be efficiently computatble using optimization methods (Proposition 5) and we give a range of possible values that this optimal reward can take (Proposition 6). We also examine what happens in the limit when $\eta T \to \infty$ (proposition 4 and 7).
>
> (1b) For discrete-time games, we show that the optimizer can always get more than the analogous continuous-time game (Proposition 2 / Proposition 8). This gap in the reward cannot be larger than $\eta T/2$ (Proposition 10) and we show a game that achieves that gap almost exactly (Proposition 9). Finally we show that under a condition, there is a large class of games that exhibits this gap between the continuous and discrete time games (Proposition 3 / Proposition 11).
>
> (2) The final result we show is a first computational lower bound for computing the optimal strategy against learners and specifically, against Fictitious play.

---

> ### Author Response · Authors · 2024-08-07
> **Answering additional questions by the reviewer**
>
> ``Why is so much exposition included in the Introduction? Is there a reason for this?''
>
> In theoretical CS papers it is customary to include an informal version of the results in the introduction. We are happy to change the format of the paper for the sake of clarity. For more details, see our response to the primary concern (of the same author) in the official rebuttal.
>
> ``The replicator dynamics is defined as a continuous time analogue of the MWU algorithm, but the MWU algorithm itself is not described until later in the paper. This can be confusing for readers, if MWU is not being used, then it should not be introduced earlier. If Replicator Dynamics needs to be described as a generalization or adaptation of MWU, MWU needs to be described first.''
>
> We defined the replicator dynamics first because it makes sense for the exposition to talk about the continuous time dynamics first, and afterwards talk about the discrete time dynamics. The replicator dynamics can be viewed as the continuous time analogue of MWU and MWU can be viewed as the discrete time analogue of the Replicator Dynamics interchangeably. We mention the MWU above, just because it is a well-known algorithm within the theory community, in order to help readers that are familiar with MWU and not with the Replicator Dynamics to make the connection. We will make it clear that the definition of MWU is not necessary in order to understand the replicator dyanmics.
>
> ``It would be helpful if n and m are described explicitly as the cardinality of the action spaces of the two players. As it is right now, it is easy to miss when they are first introduced, and later equations (e.g. Eq. 1) rely on the readers knowing what they stand for.''
>
> We will make sure to add clear references for this.
>
> ``Line 164: i1, i2 are not described in Proposition 4.''
>
> This is a typo, they are defined in assumption 1, thanks for pointing that out.
>
> ``Can you clarify the undefined notation, e.g. $\Delta(A)$?''
>
> $\Delta(\mathcal{A})$ denotes the set of all probability distributions over the set $\mathcal{A}$, i.e. $\Delta(\mathcal{A}) = \\{(x_1, x_2, \dots, x_n) |  x_1 + x_2 + \dots + x_n = 1,~ x_1\ge 0, x_2 \ge 0,\dots,x_n\ge 0 \\}$. More specifically in our setting, that is the set of all the mixed strategies for the optimizer. Similarly $\Delta(\mathcal{B})$ is the set of all mixed strategies for the learner, i.e. $\Delta(\mathcal{B}) = \\{(x_1, x_2, \dots, x_m) |  x_1 + x_2 + \dots + x_m = 1 \\}$. We will make sure to add these definitions in our paper.

---

> > ### Comment · Reviewer_jNhL · 2024-08-07
> >
> > Thanks for the detailed response to my concerns and questions, and for clarifying the significant contributions of the paper. I think a rewrite can significantly improve the paper, so I will retain my score for now. I will also go through the other reviews and rebuttals to further improve my understanding of the paper.

---

> > > ### Author Response · Authors · 2024-08-08
> > >
> > > Thank you for your prompt response.
> > > We ask if you can still consider increasing the score, if you believe in the paper's merit. This is because the other reviewers rated the readability of this paper as 3, and since we will take your comments on the writeup and incorporate them in the final revision, if this paper gets accepted.

---

### Official Review · Reviewer_dTMF · 2024-07-12

**Soundness:** 3
**Presentation:** 3
**Contribution:** 2
**Rating:** 6
**Confidence:** 4

**Summary:**

The paper studies a control problem where an optimizer plays with a mean-based learner in a zero-sum or general-sum game. The problem follows a previous line of work that aims to understand how to play optimally with no-regret learners in repeated games. The paper shows several new results. For zero-sum games, an optimal algirithm is provided for a continuous-time setting, where the learner uses a MWU-like learning dynamic. The authors show that this algorithm, when extended to the discrete-time setting, results in an algorithm that guarantees the optimizer the value of the corresponding one-shot game. For general-sum games, the paper provides a negative result, showing that no FPTAS would exist for computing the optimizer's optimal strategy.

**Strengths:**

- The problem studied follows a line of very well-motivated problems. Understanding how to play optimally with mean-based agents is very interesting.

- The paper is overall technically sound and presents some solid results.

- The zero-sum and continuous-time settings are interesting and look like reasonable choices for the problem. The paper presents non-trivial results for these settings.

**Weaknesses:**

- The informal statements of results in Section 1 could have been made clearer. The current presentation does not allow an easy comparison of results in different settings.

- The results only apply to an MWU learner, not general mean-beased learners, and it relies on the knowledge of $\eta$.

- In the discrete-time setting, the algorithm only guarantees a lower bound. There is no matching negative result.

- The negative result for the general-sum case only implies it is hard to get reward T-1, so it doesn't preclude the optimizer from obtaining sublinear regret overall. Moreover, fictitious play is a bit simplisitc as it is not no-regret. (However, I think the reduction itself is quite interesting.)

---

Minor:

- It may be clearer to state Assumption 1 as a condition or a property. Stating it as an assumption may make it sound like your result requires additional assumption, while I think the purpose of Proposition 4 is more about showing the existence of games with a utility gap.

- Line 362: closed form

**Questions:**

- Does the reduction rely on a specific tie-breaking rule, or can the tie-breaking assumption be relaxed?

- Is there any intuition why the result in Proposition 1 relies on the cardinality of BR(x)?

**Limitations:**

The authors addressed the limitations properly.

---

> ### Author Rebuttal · Authors · 2024-08-07
>
> We thank the reviewer and respond to the points raised by the review below:
>
> ``The authors show that this algorithm, when extended to the discrete-time setting, results in an algorithm that guarantees the optimizer the value of the corresponding one-shot game.''
>
> This is not exactly true. We show that for both the discrete and continuous time settings the optimizer can achieve on average more than the one-shot game. For the continuous time regime we show a closed-form solution (Theorem 3) for the rewards of the optimizer that is always more than $T \cdot Val(A)$ (Proposition 1 / Proposition 6). Moreover, in the discrete game the optimizer can always achieve more rewards than in the analogous continuous game (Proposition 2 / Proposition 8) and we show that for a large class of games the 'gap' is $\Omega(\eta T)$ (Proposition 4 / Proposition 11).
>
> ``The informal statements of results in Section 1 could have been made clearer. The current presentation does not allow an easy comparison of results in different settings.''
>
> We will rewrite the statements more clearly and compare them to previous related work in the new revision.
>
> ``The results only apply to an MWU learner, not general mean-beased learners, and it relies on the knowledge of $\eta$''
>
> We believe that our result could be extended to the case where $\eta$ is not known in advance -- by an algorithm that learns $\eta$ on-the-go. This was not the focus of our paper and it is left for future work. For optimizing against arbitrary mean-based learners, we do not know if there is a clean solution, and devising an optimal strategy against any mean-based learner might be a very hard research problem (as also discussed by Brown et al., ``Is Learning in Games Good for the Learners?'')
>
> ``In the discrete-time setting, the algorithm only guarantees a lower bound. There is no matching negative result.''
>
> We do provide upper and lower bounds, which are tight up to constant factors.
> Proposition 4 proves that in discrete games, the optimizer cannot obtain more than $\eta T/2$ reward compared to the continuous time analogue (Proposition 3 / Proposition 10). For the positive result, we show that there exists a game where this gain is optimal up to a multiplicative $1+o(1)$ (Proposition 3 / Proposition 9) factor. Furthermore, for a class of games satisfying a mild condition, we show that the optimizer can gain $\Omega(\eta T)$ more reward than the Value of the game in the discrete-time setting (Proposition 4).
>
> ``The negative result for the general-sum case only implies it is hard to get reward T-1, so it doesn't preclude the optimizer from obtaining sublinear regret overall. Moreover, fictitious play is a bit simplisitc as it is not no-regret. (However, I think the reduction itself is quite interesting.)''
>
> Both of your observations are correct; our reduction does not prove hardness of approximation of the optimal rewards, and also it is based on the assumption that the learner employs fictitious play (where $\eta \to \infty$). However, we see this lower bound as a first step to possibly unlocking the mystery of the strategization against MWU, that was first introduced by [1] 5 years ago. Many other works (inclduing [2], [3]) have this left as an open problem, and we believe that this problem is computationally hard. Our hardness result is the first in this line of work and we are hopeful that in the future we will be able to prove hardness of approximation and relax the $\eta \to \infty$ assumption.
>
> ``It may be clearer to state Assumption 1 as a condition or a property. Stating it as an assumption may make it sound like your result requires additional assumption, while I think the purpose of Proposition 4 is more about showing the existence of games with a utility gap.''
>
> You are right, the point is to show that a certain category of games have a utiity gap between their discrete and continuous time analogues. We will phrase the assumption as a condition instead, for the utility gap to exist.
>
> ``Line 362: closed form''
>
> Thanks for pointing the typo out.
>
> ``Does the reduction rely on a specific tie-breaking rule, or can the tie-breaking assumption be relaxed?''
>
> Yes, the reduction does rely on the specific tie breaking rule, that is only used however in the first round of the game where the historical rewards of the learner for each action is 0. We did not see a simple way to relax this tie breaking rule, and we leave relaxing this assumption for future work.
>
> ``Is there any intuition why the result in Proposition 1 relies on the cardinality of BR(x)?''
>
> The less best responses for the learner there are the longer it will take for the learner to converge from the uniform distribution to that best response. Imagine a game where every action of the learner is a best response; in that scenario the rewards for the optimizer will be exactly $T \cdot Val(A)$. In comparison, in the case where there is only a single best response, the optimizer will gain extra utility because of the time it takes for the learner to learn that best response. The less best responses, the more time it takes to concentrate all the probability for the learner in the best responses and therefore the more utility for the optimizer.

---

> > ### Comment · Reviewer_dTMF · 2024-08-12
> >
> > Thank you. I appreciate the detailed responses and answers, which are very helpful for understanding the results. I'm now more positive about the paper.

---

### Official Review · Reviewer_bLvs · 2024-07-13

**Soundness:** 3
**Presentation:** 3
**Contribution:** 3
**Rating:** 7
**Confidence:** 3

**Summary:**

This paper outlines conditions under which, in repeated two-player zero-sum and general-sum games between a learner with an online learning strategy and an optimizer that knows the learner's strategy and utility function, the optimizer can learn a policy to achieve a higher average utility than the value of the one-shot game. For the two-player zero-sum game setting, assuming the learner selects actions following Replicator Dynamics in continuous time, this paper proposes an algorithm in which the optimizer can take a constant (over time) optimal strategy that provably maximizes its utility. In discrete time with the learner following a Multiplicative Weights Update (MWU) strategy, the continuous time optimal utility of the optimizer is shown to lower bound the discrete time optimal performance following the same strategy. For general-sum games, this paper proves a computational hardness result showing it is NP-hard to approximate the optimal utility for the optimizer against a Best Response learner. This is proven via a reduction from the Hamiltonian cycle problem.

**Strengths:**

1. The paper does a good job at highlighting the motivating questions, and the main ideas of the proposed approach are in general clearly explained. The proof for the reduction from the Hamiltonian cycle was quite intuitively explained too, and I really appreciate it.

2. The theoretical contributions seem sound and relevant to the optimizer-learner setting in two-player zero-sum and general-sum games. Please note that I have not thoroughly checked the proof details in the appendix.

**Weaknesses:**

1. This paper primarily focuses on theoretical bounds on optimal utility and computational hardness guarantees for learning an optimal strategy for the optimizer against mean-based learners in the zero-sum and general-sum games. But there are no empirical evaluations of the proposed algorithms. In section 1.2 Related Work, authors point out prior work in contracts and auction design, which could also be broadly categorized as mechanism design. Some recent related papers (eg. [1],[2] ) have proposed experimental frameworks to analyze interactions in similar optimizer-learner frameworks, which could be referred to for similar experiment design with repeated games and bandit agents.

[1] Guo, W., Agrawal, K.K., Grover, A., Muthukumar, V.K. and Pananjady, A., 2022, March. Learning from an Exploring Demonstrator: Optimal Reward Estimation for Bandits. In International Conference on Artificial Intelligence and Statistics (AISTATS).

[2] Banerjee, A., Phade, S.R., Ermon, S. and Zheng, S., MERMAIDE: Learning to Align Learners using Model-Based Meta-Learning. Transactions on Machine Learning Research.

2. The connection to optimal control could perhaps be better motivated and explained.

3. Confusing notation that can be improved:
- Line 255 and 256: Should it be $R_{cont}(x, h(0), T, A, -A)$ and $R^*_{cont}(h(0),T,A,-A)$?
- Line 308: "for each node $v_i$ of the graph, the learner has two associated actions $v_i$ and $v_i'$" - the overloaded use of $v_i$ is confusing.
- I might have missed it, but what is the relation between $h_i(t)$ and $h(t)$? Is it explicitly defined somewhere in the paper?

**Questions:**

1. line 164: "$i_1, i_2$ are defined in Proposition 4. - Should this instead be  "$b_{i_1}, b_{i_2}$ defined in Assumption 1" ?

2. If I understand correctly, the reason why "the learner would have to change actions frequently" (line 153) for the optimizer to get a higher optimal utility, is so that the optimizer can exploit the gain by playing an optimal strategy at every time step? In this paper, for the discrete time dynamics, the learner's strategy is therefore assumed stochastic to ensure that the learner frequently changes actions. What would be the effect of assuming a stochastic learner in the continuous time case?

**Limitations:**

Yes, authors have discussed the limitations and potential future directions for their approach.

---

> ### Author Rebuttal · Authors · 2024-08-07
>
> We thank  the reviewer and respond to the points raised by the review below:
>
> ``In section 1.2 Related Work, authors point out prior work in contracts and auction design, which could also be broadly categorized as mechanism design. Some recent related papers (eg. [1],[2] ) have proposed experimental frameworks to analyze interactions in similar optimizer-learner frameworks, which could be referred to for similar experiment design with repeated games and bandit agents.''
>
> Thank you for pointing out these papers. We will include them in the related work section in the new revision of our paper.
>
> ``The connection to optimal control could perhaps be better motivated and explained.''
>
> The optimal control paragraph is explaining how Theorem 1 can be derived using the HJB equation. The problem of strategizing against MWU can be viewed as a control problem where the system has some dynamics (in this case the dynamics of MWU), a utility function (the rewards of the optimizer), and the control which is the strategy of the optimizer. The HJB equation gives a partial differential equation that when solved gives the closed-form solution of Theorem 1. We included it because we think it might be helpful for thinking about the general sum case. The connection of optimizing against learners and control has already been discussed in prior work [24], yet from a different angle and they did not discuss the HJB equation. We will definitely include a more detailed explanation in the new revision of the paper.
>
> ``Line 255 and 256: Should it be $R_{cont}(x, h(0), T, A, -A)$ and $R_{cont}^*(h(0), T, A, -A)$?''
>
> You are right, this is a typo. We will fix it in the new revision.
>
> ``Line 308: "for each node $v_i$ of the graph, the learner has two associated actions $v_i$ and $v_i'$" - the overloaded use of $v_i$ is confusing.''
>
> We will use a different notation for the actions, thank you for pointing this out.
>
> ``I might have missed it, but what is the relation between $h_i(t)$ and $h(t)$? Is it explicitly defined somewhere in the paper?''
>
> $h(t)$ is an $m$-dimensional vector and thus $h_i(t)$ is the $i$-th coordinate of that vector. We will make sure to include it in the definitions.
>
> ``Line 164: $i_1,i_2$ are defined in Proposition 4. - Should this instead be $b_1, b_2$ defined in Assumption 1" ?''
>
> You are right, this is a typo. It should be " ... where $i_1, i_2$ are defined  in Assumption 1 ...", not in Proposition 4.
>
> ``If I understand correctly, the reason why "the learner would have to change actions frequently" (line 153) for the optimizer to get a higher optimal utility, is so that the optimizer can exploit the gain by playing an optimal strategy at every time step? In this paper, for the discrete time dynamics, the learner's strategy is therefore assumed stochastic to ensure that the learner frequently changes actions. What would be the effect of assuming a stochastic learner in the continuous time case?''
>
> First of all there is a typo in that line (153). It should be "the optimizer would have to change actions frequently". Both in the discrete and the continuous dynamics the learner and the optimizer are allowed to play stochastically. To better understand what we mean by the optimizer having to change actions frequently take a look at Proposition 9, which exemplifies the main difference between the continuous and discrete time dynamics. It studies the game of Matching Pennies, both in  discrete and continuous-time. For the continuous-time dynamics, the best value the optimizer can get is $0$. This is a consequence of Theorem 3. However, for the discrete time dynamics, it is possible to get $\Omega(\eta T)$ reward for the optimizer. This is achieved when the optimizer switches actions all the time, and then it is possible to gain positive reward, as presented in the proof. Essentially, the main difference between the continuous time dynamics and the discrete time dynamics is that in the continuous case the learner changes the strategy smoothly and therefore cannot be exploited as we can exploit the discrete time dynamics. Intuitively, the learner is ``slower to respond'' in the discrete setting, hence a frequent change of actions from the optimizer could benefit the latter.

---

> > ### Comment · Reviewer_bLvs · 2024-08-13
> > **Acknowledgement**
> >
> > Thank you for the response and clarifications. I have also read the other reviews and responses, which helped improve my understanding of this work. I will maintain the current score.

---

### Decision · Program_Chairs · 2024-09-25

**Decision:**

Accept (poster)

**Comment:**

The paper examines a two-player repeated game setting involving two types of players: one is an online learner, while the other is a planner who accounts for the learner's behavior when optimizing their strategy. Overall, the reviewers found the problem studied and the characterization results concerning the exploitability of a mean-based learner interesting, and the corresponding analysis sound. However, concerns were raised about the quality of the presentation. While the format is not atypical for a theoretical paper, there is room for improvement. The authors are strongly encouraged to address these concerns and revise their paper according to the suggested edits in their rebuttal.